# CRITIQUE-GUIDED DISTILLATION FOR EFFICIENT AND ROBUST LANGUAGE MODEL REASONING

**Berkcan Kapusuzoglu, Supriyo Chakraborty, Chia-Hsuan Lee, Sambit Sahu**
Capital One
McLean, VA 22102, USA
{berkcan.kapusuzoglu}@capitalone.com

## ABSTRACT

Supervised fine-tuning (SFT) with expert demonstrations often suffers from the imitation problem, where models reproduce correct responses without internalizing the underlying reasoning. We propose CRITIQUE-GUIDED DISTILLATION (CGD), a multi-stage training framework that augments SFT with teacher-generated *explanatory critiques* and *refined responses*. Instead of directly imitating teacher outputs, a student learns to map the triplet of prompt, its own initial response, and teacher critique into the refined teacher response, thereby capturing both *what* to output and *why*. On mathematical reasoning benchmarks, CGD achieves substantial gains across LLaMA and Qwen families: +15.0% on AMC23 and +12.2% on MATH-500 over CFT, while avoiding the format drift that plagues critique-based methods. Cross-family validation on Qwen2.5-Math-7B with diverse teachers (Claude Sonnet 3.7 to weaker open-source models) achieves state-of-the-art performance (50.4 avg, +22.6% over base) with 144× less compute than RL methods. Critically, despite training on data containing no code, CGD generalizes to out-of-distribution benchmarks: +4.88% on HumanEval (code generation), and preserved or improved performance on GPQA, MUSR, TruthfulQA, and BBH, while CFT suffers catastrophic forgetting (-21.3% on IFEval). These results establish CGD as a cost-effective intermediate training paradigm that can serve as a warm-start before reasoning SFT or RL, offering a scalable enhancement to modern LLM training workflows.

## 1 INTRODUCTION

Supervised fine-tuning (SFT) is a foundational technique for teaching large language models (LLMs) to perform diverse downstream tasks by mimicking expert-annotated outputs (Wei et al., 2022; Sanh et al., 2022). Despite its success, vanilla SFT has notable limitations: it increases model's tendency to hallucinate (Gekhman et al., 2024), exhibits limited out-of-distribution generalization (Chu et al., 2025), and struggles to generalize to harder problem instances (Sun et al., 2024; 2025). These shortcomings raise fundamental questions about SFT's capacity for robust, and complex reasoning.

An alternative approach to improve reasoning leverages *critique* and *revision* at inference time: a model generates an initial answer, critiques it, then refines its output based on that critique (Kim et al., 2023; Madaan et al., 2023; Shinn et al., 2023; Saunders et al., 2022). While effective, these multi-pass prompting methods incur high computational costs and latency during deployment.

To integrate critique signals without extra inference costs, recent works has moved these steps into training. Rejection Sampling Fine-Tuning (RFT) (Yuan et al., 2023) trains the model on its own generated outputs that are verified or ranked by a reward model, thus incorporating value-based feedback. Critique Fine-Tuning (CFT) instead trains a student model to reproduce teacher-generated critiques (Wang et al., 2025). Although CFT outperforms vanilla SFT on several math benchmarks, prolonged CFT can induce output-format drift, overfitting to critique patterns rather than stable answer structures and its gains are sensitive to the quality of the critiques provided.

In this work, we introduce CRITIQUE-GUIDED DISTILLATION (CGD) (Fig. 1), a novel multi-step fine-tuning paradigm in which a student model learns to transform its own initial outputs into high-quality refinements, rather than just generating critiques. Concretely, we condition a teacher model to produce both critiques and corresponding corrected answers, and train the student to map its raw response to the teacher's refined version by conditioning it on the critique. By internalizing not only how to identify errors but also what a polished response looks like, CGD closes the loop between diagnosis and correction. Importantly, the use of initial answers and critiques is restricted to the training phase: at inference time CGD requires only the original prompt and produces the refined answer in a single pass, with no need for critiques. This design both avoids format drift compared to CFT (Fig. 1) and eliminates the inference-time overhead of multi-pass critique methods.

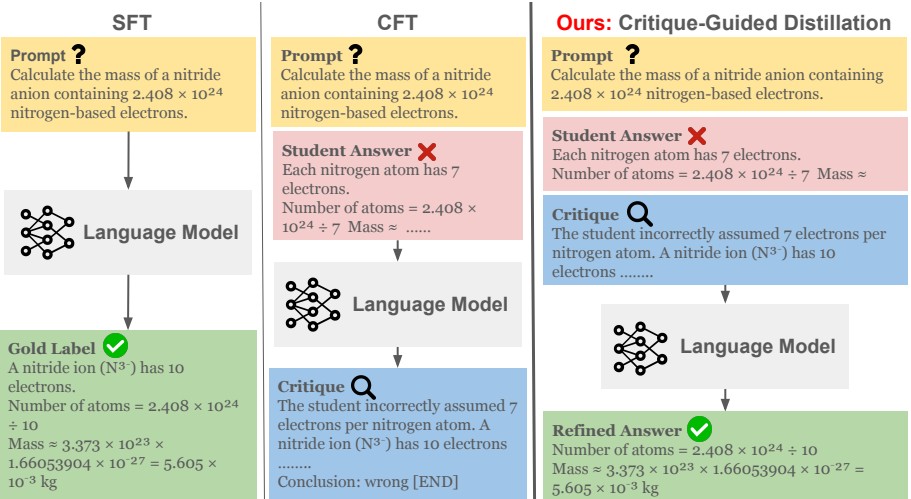

Figure 1: **Comparing Supervised fine-tuning (SFT), Critique Fine-Tuning (CFT) and CRITIQUE-GUIDED DISTILLATION (CGD)**. Unlike CFT, which trains the student to generate critiques, CGD conditions training on both the initial answer and critique but at test time generates the final answer directly in a single pass.

By conditioning answer generation on the critique, CGD avoids format drift (the model continues to generate answers, not critiques) and ensures feedback is explicit and grounded. Critically, while CFT suffers from catastrophic forgetting (-21.3% drop on IFEval instruction-following), CGD preserves general capabilities while improving reasoning. We empirically validate CGD on mathematical reasoning and broad knowledge benchmarks, observing +17.5% and +15.0% absolute accuracy gains over SFT and CFT respectively on the challenging AMC23 dataset for `LLaMA3.1-8B Instruct`. Similarly, on OlympiadBench, CGD achieves a +12.9% gain over SFT and +8.0% over CFT. For `S1.1-3B`, CGD achieves +12.2% and +7.5% gains over CFT on MATH-500 and AMC23, respectively. Figure 2 visualizes this performance trend, showing that CGD consistently improves over strong baselines such as Distilled SFT and CFT across all evaluation tasks. In contrast to prior methods, CGD explicitly conditions the student on the input prompt, its own initial answer, and the teacher's critique, enabling the model to internalize not just what the correct refinement is but why the refinement is needed. This richer supervision leads to more robust and generalizable reasoning behavior.

In summary, our contributions are as follows:

- We introduce CRITIQUE-GUIDED DISTILLATION (CGD), a novel and efficient fine-tuning framework that trains a student model on the full cycle of self-correction: from a flawed initial answer, to a critique, to a refined output.

- We demonstrate that CGD achieves state-of-the-art performance, outperforming strong distillation and critique-based baselines. On `LLaMA3.1-8B Instruct` and `S1.1-3B`, CGD achieves absolute gains of **+5.4%** and **+7.2%** over CFT on math reasoning benchmarks with particularly strong improvements on challenging tasks (+15.0% AMC23, +8.0% OlympiadBench). Cross-family validation on `Qwen2.5-Math-7B` (using diverse teachers from Claude Sonnet 3.7 to weaker open-source models) achieves **+22.6% gain**

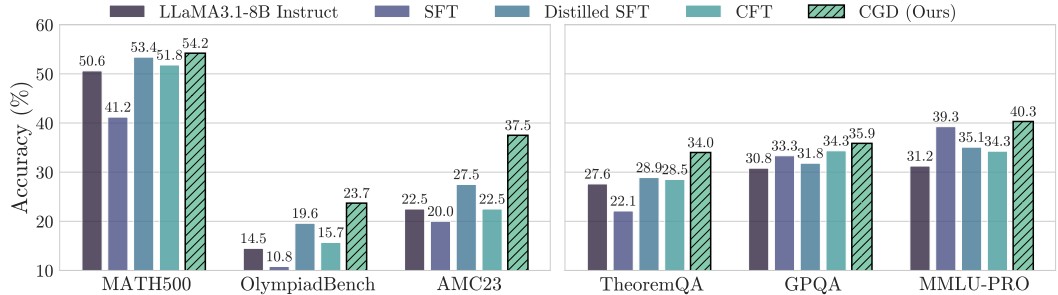

Figure 2: **Performance comparison of CGD, using a `LLaMA3.3-70B Instruct` teacher model to generate critiques and refined answers, with 100K samples from WebInstruct (Yue et al., 2024).** The `LLaMA3.1-8B Instruct` student model is trained using the input prompt, initial answer and the the critique as input, and the refined answer as the target. Baselines include Distilled SFT, which uses only the input prompt as input to imitate refined answers from the same teacher (`LLaMA3.3-70B Instruct`) model on the same WebInstruct data, and CFT, which trains on GPT-4o-generated critiques (Wang et al., 2025).

**over base** (27.8 → 50.4) and state-of-the-art results with **144× less compute than reinforcement learning (RL) methods** (8 vs 1152 GPU-hours). This positions CGD as a cost-effective intermediate training paradigm that can precede reasoning SFT or RL in modern training pipelines.

- Despite training on data containing no code (WebInstruct spans Math, Physics, Chemistry, and Business, but excludes code), CGD transfers to OOD benchmarks including HumanEval (+4.88% pass@1 on code generation), while preserving or improving performance on GPQA, MUSR, TruthfulQA, and BBH. By contrast, CFT suffers catastrophic forgetting (-21.3% on IFEval), demonstrating CGD's unique advantage in learning broadly transferable reasoning skills beyond domain-specific patterns.

- We conduct thorough ablation studies across model families (LLaMA (Grattafiori et al., 2024), Qwen/S1.1 (Muennighoff et al., 2025), Mixtral (Jiang et al., 2024), OLMo (Groeneveld et al., 2024)), training datasets (WebInstruct (Yue et al., 2024), MetaMathQA (Yu et al., 2024a)), and diagnostic analyses (Appendix C), proving CGD's robustness. Our method is significantly more stable to hyperparameter changes than CFT and produces models with improved confidence and reasoning quality.

## 2 RELATED WORK

**Supervised Fine-Tuning Limitations.** Standard supervised fine-tuning (SFT) trains Large Language Models (LLMs) to mimic expert demonstrations, but it often induces the imitation problem, where models reproduce outputs without internalizing reasoning processes. Previous studies show that fine-tuning language models on new knowledge increases model's tendency to hallucinate (Gekhman et al., 2024). Furthermore, fine-tuned models exhibit poor out-of-distribution performance (Chu et al., 2025), and gains on familiar data often come at the cost of reliability on unseen distributions (Li et al., 2025).

Recent work has also shown that SFT on reasoning trajectories can substantially boost mathematical problem-solving with only a few thousand examples (Muennighoff et al., 2025; Ye et al., 2025). Nonetheless, vanilla SFT still struggles to generalize to harder problem instances, leaving open the question of its limits on complex reasoning (Sun et al., 2024; 2025). These limitations motivate integrating critique-and-correction mechanisms beyond naive answer imitation to achieve more robust reasoning and improved downstream performance.

**Distillation from Reasoning Traces and Self-Correction.** Knowledge distillation from large teachers has proven effective for improving student reasoning (Hinton et al., 2015). ORCA (Mukherjee et al., 2023) and subsequent work (Mitra et al., 2024) distill reasoning by training students on detailed explanation traces (step-by-step CoT) generated by GPT-4. MoDE-CoTD (Li et al., 2024)

advances this using mixture of LoRA experts for complex reasoning. While effective, these methods distill the teacher's *reasoning process* for solving problems from scratch. In contrast, CGD trains students to *correct their own errors*, learning from student mistakes rather than imitating teacher traces. This conditions on actual failure modes, aligning correction with student-specific weaknesses.

Prior work has also shown that LLMs can critique their own outputs and refine them based on self-generated feedback (Kim et al., 2023; Madaan et al., 2023; Saunders et al., 2022). However, these methods depend on multi-pass prompting and incur substantial inference-time overhead. To address this, several fine-tuning approaches endow smaller models with self-correction capabilities (Shridhar et al., 2023; Yu et al., 2024b), but still require separate critique and refinement passes at inference.

More recently, Critique Fine-Tuning (CFT) (Wang et al., 2025) trains students to generate teacher critiques, enabling single-pass generation. However, CFT focuses on producing critique tokens, causing output-format drift. In contrast, CGD directly fine-tunes on the *refined answer*, preserving output consistency while retaining critique-driven improvements and outperforming CFT on reasoning tasks.

**Feedback Quality and Mechanisms for Self-Refinement.** The quality and specificity of feedback critically determines learning effectiveness. Education research emphasizes that feedback must be actionable and specific rather than generic to drive meaningful improvement (Borges et al., 2023). Building on these principles, ELAD (Zhang et al., 2024) actively selects high-uncertainty examples via explanation-step uncertainties to provide targeted supervision, reducing annotation costs while preserving student performance. Similarly, work on tutorial systems demonstrates that real-time explanatory feedback to human tutors improves learning outcomes (Lin et al., 2023), while sequence labeling approaches highlight desired versus undesired response components for targeted improvement (Lin et al., 2024). These findings collectively underscore that explanation-rich feedback enables models to learn both *what* is correct and *why*, motivating CGD's explicit incorporation of critiques as conditioning signals.

Beyond feedback quality, various works study different granularities and mechanisms for self-refinement. DeCRIM breaks high-level instructions into fine-grained constraints to guide targeted corrections (Ferraz et al., 2024), while LLMRefine leverages human-defined error categories to produce pinpointed feedback (Xu et al., 2024; Paul et al., 2024). DCR further modularizes this pipeline by separating error detection, critique generation, and final refinement into distinct stages (Wadhwa et al., 2024). Other approaches enhance correction accuracy by incorporating external tools—code executors for programming (Chen et al., 2023a; 2024), proof assistants for mathematics (First et al., 2023), and search engines for factual validation (Gao et al., 2023; Gou et al., 2024). In contrast, our method relies solely on a large teacher LLM to provide general-purpose critiques, avoiding task-specific external signals while maintaining broad applicability.

## 3 CRITIQUE-GUIDED DISTILLATION (CGD)

In this section we describe CGD and provide analysis of its training procedure.

### 3.1 OVERVIEW

The key intuition behind CGD is to train a student model to perform a complete reasoning loop: from generating an initial student answer, to understanding a critique of that answer, to producing a final, refined output. By internalizing not only how to identify its own errors but also how to correct them, the student learns a more robust and generalizable reasoning process. This approach, summarized in Figure 3, proceeds in three main stages:

1. **Student Answer Generation:** The student baseline $S_{\theta_{\text{init}}}$ produces a noisy response $y' \sim S_{\theta_{\text{init}}}(\cdot|x)$.

2. **Critique Generation:** The teacher model $T_\phi$ critiques this response, generating a textual explanation of its flaws or merits, $c \sim T_\phi(\cdot|x, y')$.

3. **Refined Answer Generation:** The teacher produces a gold-standard, refined answer $\hat{y} \sim T_\phi(\cdot|x, y', c)$, conditioned on all prior context.

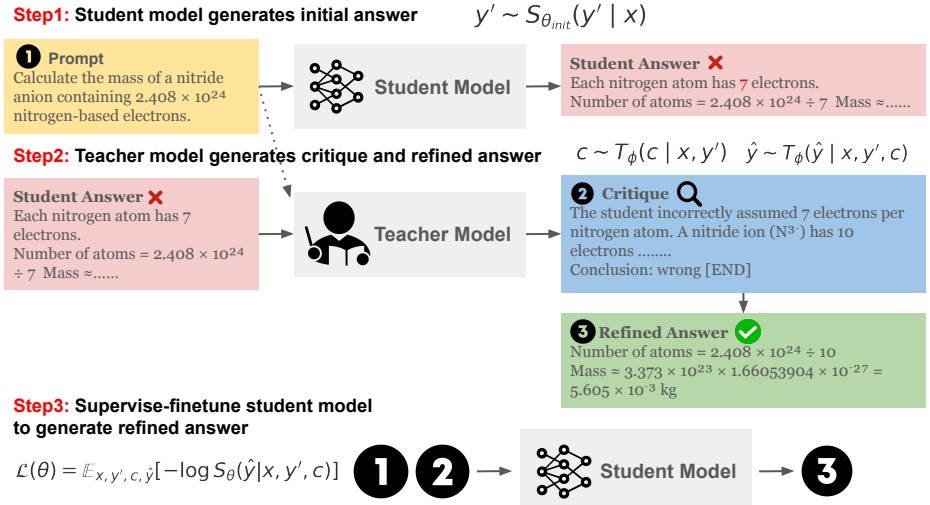

Figure 3: **Overview of CRITIQUE-GUIDED DISTILLATION (CGD)**. During training, the student produces an initial response, the teacher supplies a critique and refined answer, and the student is fine-tuned to map from (prompt, student answer, critique) → refined answer. At inference, however, only the prompt is provided, and the student directly outputs the refined answer in one pass.

This training-time-only intervention ensures that feedback is explicit and grounded. By conditioning the final answer generation on the critique, CGD avoids the format drift seen in methods like CFT, as the model's objective remains to generate answers, not critiques.

## 3.2 TRAINING OBJECTIVE

The student is fine-tuned on the augmented dataset $((x, y', c), \hat{y})$ using a standard language modeling objective. As summarized in Algorithm 1, the goal is to minimize the negative log-likelihood of the teacher's refined answer, conditioned on the full context:

$$\mathcal{L}(\theta) = \mathbb{E}_{(x,y',c,\hat{y})}[-\log S_\theta(\hat{y} \mid x, y', c)]. \qquad (1)$$

Crucially, at inference time, CGD requires only a single forward pass, making it identical in computational cost to standard SFT.

---

**Algorithm 1** CRITIQUE-GUIDED DISTILLATION (CGD)

---

1: **Input:** Dataset $\mathcal{D} = \{x_i, y_i\}_{i=1}^N$, Student $S_{\theta_{init}}$, Teacher $T_\phi$
2: **Output:** Fine-tuned student $S_\theta$
3: Initialize augmented dataset $\mathcal{D}' \leftarrow \emptyset$
4: **for** each $x_i \in \mathcal{D}$ **do**
5:   Generate student answer: $y_i' \sim S_{\theta_{init}}(y|x_i)$
6:   Generate critique: $c_i \sim T_\phi(c|x_i, y_i')$
7:   Generate refined answer: $\hat{y}_i \sim T_\phi(\hat{y}|x_i, y_i', c_i)$
8:   $\mathcal{D}' \leftarrow \mathcal{D}' \cup \{(x_i, y_i', c_i, \hat{y}_i)\}$
9: **end for**
10: Train $S_\theta$ on $\mathcal{D}'$ by minimizing $\mathcal{L}(\theta)$
11: **return** $S_\theta$

---

From a probabilistic perspective, the CGD framework can be interpreted as a form of Bayesian inference. The student's initial output distribution, $P(y|x)$, acts as a prior belief. The critique $c$ serves as new evidence. The goal of the student is to learn the posterior distribution $P(y|x, c)$, which is proportional to the likelihood of the critique given a refined answer, $P(c|x, y)$, multiplied by the prior. By training the student to match the teacher's refined answer $\hat{y}$ (which is drawn from a high-quality posterior), CGD systematically reduces the model's uncertainty in line with the critique's guidance, formalizing why it produces more confident and accurate predictions (see Section 4.3.2).

# 4 EXPERIMENTS

Our experiments are designed to show that CRITIQUE-GUIDED DISTILLATION (CGD) is a highly efficient and effective method for improving the reasoning capabilities of LLMs. We demonstrate that CGD significantly outperforms strong fine-tuning baselines, including standard SFT, Distilled SFT, and CFT, across a diverse suite of challenging math and reasoning benchmarks. Furthermore, we show that CGD is dramatically more compute-efficient than recent RL methods and exhibits superior robustness to hyperparameter choices compared to other critique-based techniques. This efficient learning of a robust self-correction skill is the mechanism that directly contributes to the superior performance on downstream benchmarks.

## 4.1 EXPERIMENTAL SETUP

### 4.1.1 DATASETS

We consider two datasets for training: WebInstruct (Yue et al., 2024), and MetaMathQA (Yu et al., 2024a). WebInstruct is a web-crawled instruction dataset that spans a wide range of topics, including Math, Physics, Chemistry, and more. MetaMathQA is a dataset based on GSM8K (Cobbe et al., 2021b) and MATH (Hendrycks et al., 2021) which synthesizes more questions and answers by rephrasing and other augmentation techniques. We randomly sample 100k examples from each dataset as training data.

We evaluate on two sets of benchmarks capturing both mathematical reasoning and broader STEM-oriented problem solving. **Group 1: Math Reasoning** comprises MATH500 (Hendrycks et al., 2021), Minerva-Math (Lewkowycz et al., 2022), GSM8K (Cobbe et al., 2021a), Olympiad-Bench (He et al., 2024), and AMC23. **Group 2: General Reasoning** includes TheoremQA (Chen et al., 2023b), GPQA (Rein et al., 2023), and MMLU-Pro (Wang et al., 2024).

To evaluate model capabilities beyond math and science reasoning, we also report results on the following datasets to evaluate general instruction-following and question answering abilities: IFEval (Zhou et al., 2023), MUSR (Sprague et al., 2024), TruthfulQA (Lin et al., 2022), and BIG-Bench Hard (BBH) (Suzgun et al., 2022).

### 4.1.2 BASELINE AND TRAINING SETTINGS

We evaluate CGD across two student–teacher pairs to test both within-family and cross-family robustness:

- **LLaMA family:** `LLaMA3.1-8B Instruct` as the *student model*, and `LLaMA3.3-70B Instruct` as the *teacher model*.

- **Qwen family:** `S1.1-3B`[1] as the *student model*, and `S1.1-32B`[2] as the *teacher model*.

We compare CGD to three supervised fine-tuning baselines: (i) **Standard SFT:** fine-tunes the student model to generate gold answers conditioned only on the input prompt. (ii) **Distilled SFT:** fine-tunes the student to reproduce the teacher's refined answers, where each refinement is obtained by prompting the teacher with the input prompt, the student's initial answer, and the teacher-generated critique. (iii) **Critique Fine-Tuning (CFT):** fine-tunes the student to generate the teacher-provided critiques conditioned on the input prompt and the student's initial answer.[3]

All experiments are trained on 16 Nvidia A100 GPUs for 30 minutes, amounting to a total of 8 A100 GPU-hours per experiment, using identical data splits and hyperparameters across methods (see Appendix A for more details).

---

[1]https://huggingface.co/simplescaling/s1.1-3B

[2]https://huggingface.co/simplescaling/s1.1-32B

[3]Due to licensing and regulatory restrictions, we were unable to directly use certain models (e.g., Qwen) as students or pair GPT-4o as a teacher. Accordingly, our experiments focus on the LLaMA and S1.1 families, where such usage is permitted.

## 4.2 MAIN RESULTS

We report the evaluation results of training on WebInstruct in Table 1. We evaluate two student–teacher pairs: `LLaMA3.1-8B Instruct` with `LLaMA3.3-70B Instruct` as the teacher, and `S1.1-3B` with `S1.1-32B` as the teacher. In addition to our CFT experiments using `LLaMA3.3-70B Instruct`, we also include a variant of CFT that uses 50K examples distilled with GPT-4o, sourced from Wang et al. (2025) (denoted as *CFT* with GPT-4o*). Cross-family validation on `Qwen2.5-Math-7B` is presented in Section 4.3. A full breakdown of additional results, including ablation studies on different model architectures, teacher models, on math-specific training data (MetaMathQA), hyperparameter sensitivity, and critique composition, is provided in Appendix B

Table 1: **Evaluation of fine-tuning methods on `LLaMA3.1-8B Instruct` and `S1.1-3B`.** Results are reported across math-focused (Group 1) and general reasoning (Group 2) benchmarks using WebInstruct as training set (100K samples). CGD consistently achieves the best average performance across both families. Bold: best, underline: 2nd. Δ rows show CGD improvement over CFT. Cross-family validation on Qwen2.5-Math-7B is presented separately in Table 3.

| Method | MATH500 | Math Reasoning Tasks (Group 1) | | | | | General Reasoning Tasks (Group 2) | | | |
| | | Minerva-Math | GSM8K | OlympiadBench | AMC23 | Avg. | TheoremQA | GPQA | MMLU-PRO | Avg. |
|---|---|---|---|---|---|---|---|---|---|---|
| LLaMA3.1-8B Instruct | 50.6 | _33.5_ | 85.3 | 14.5 | 22.5 | 41.3 | 27.6 | 30.8 | 31.2 | 29.9 |
| + SFT | 41.2 | 24.6 | 80.7 | 10.8 | 20.0 | 35.5 | 22.1 | 33.3 | 39.3 | 31.6 |
| + Distilled SFT | 53.4 | 32.7 | 85.3 | _19.6_ | _27.5_ | _43.7_ | _28.9_ | 31.8 | 35.1 | 31.9 |
| + CFT* with GPT-4o | **54.8** | 33.1 | **86.2** | 18.2 | 25.0 | 43.5 | **35.0** | 30.3 | **40.8** | _36.4_ |
| + CFT | 51.8 | 32.7 | 84.8 | 15.7 | 22.5 | 41.5 | 28.2 | _34.3_ | 34.2 | 32.4 |
| + CGD | _54.2_ | **33.6** | _85.7_ | **23.7** | **37.5** | **46.9** | _34.0_ | **35.9** | 40.3 | **36.7** |
| Δ = CGD - CFT | 2.4 | 0.9 | 0.9 | 8.0 | 15.0 | 5.4 | 5.8 | 1.6 | 6.1 | 4.3 |
| S1.1-3B | 54.0 | 16.9 | 76.8 | _20.6_ | _30.0_ | 35.4 | 21.6 | 16.7 | 13.7 | 17.9 |
| + SFT | 55.4 | 18.8 | 76.8 | 19.6 | _30.0_ | 40.1 | 22.8 | _29.8_ | **36.9** | 29.8 |
| + Distilled SFT | _60.6_ | _22.1_ | **83.1** | 20.4 | 22.5 | _41.7_ | **34.9** | 29.3 | 36.4 | **33.5** |
| + CFT | 49.6 | 21.0 | 77.3 | 19.3 | 27.5 | 38.9 | 25.9 | 26.7 | 35.9 | 29.5 |
| + CGD | **61.8** | **27.9** | _82.5_ | **23.1** | **35.0** | **46.1** | _32.8_ | **31.8** | 35.7 | _33.4_ |
| Δ = CGD - CFT | 12.2 | 6.9 | 5.2 | 3.8 | 7.5 | 7.2 | 5.6 | 5.0 | -0.2 | 3.5 |

CGD consistently improves over CFT across both families. On `LLaMA3.1-8B`, CGD achieves +5.4 average gain on math reasoning (Group 1) and +4.3 on general reasoning (Group 2), with particularly strong improvements on OlympiadBench (+8.0) and AMC23 (+15.0). On `S1.1-3B`, CGD achieves even larger gains of +7.2 on math and +3.5 on general reasoning, including notable improvements on MATH500 (+12.2), Minerva-Math (+6.9), and AMC23 (+7.5). These results demonstrate that critique-guided training enhances reasoning ability more broadly than CFT and distilled SFT across diverse student–teacher settings.

We additionally evaluate on out-of-distribution benchmarks (Table 2). CGD matches or surpasses all baselines, confirming that critique-conditioned training preserves or improves humanities, logic, factual QA, instruction-following and question-answering abilities. By contrast, CFT's IFEval accuracy falls from 76.6% to 55.6%, likely because CFT is optimized to predict critiques rather than final answers, which is an objective that can disrupt format-sensitive instruction following.

**Code Generation (HumanEval).** To validate domain-agnostic generalization, we evaluated `LLaMA3.1-8B Instruct` models on HumanEval (Chen et al., 2021) (Python code generation) zero-shot, without any code training data. CGD achieves +4.88% Pass@1 improvement (59.75% → 64.63%) over `LLaMA3.1-8B Instruct`, and outperforming CFT (+4.27%). This demonstrates CGD's self-correction transfers to structured generation tasks, confirming it learns generalizable reasoning patterns beyond domain-specific heuristics.

Table 2: **Effect of different fine-tuning strategies on `LLaMA3.1-8B Instruct` across diverse benchmarks.** While CGD preserve or improve performance, CFT severely degrades general capabilities.

| Method | IFEval | MUSR | TruthfulQA | BBH |
|---|---|---|---|---|
| LLaMA3.1-8B Instruct | 76.9 | 37.8 | 54.0 | **48.3** |
| + SFT | 76.6 | 36.9 | 52.0 | 48.0 |
| + Distilled SFT | **77.5** | 39.0 | 53.9 | 47.0 |
| + CFT* w/ GPT4o | 55.6 | 35.0 | 53.5 | 44.2 |
| + CGD | 76.1 | **39.3** | **54.5** | 47.1 |

### 4.3 CROSS-FAMILY VALIDATION ON QWEN2.5-7B-MATH

To further validate the robustness and cross-family effectiveness of our approach, we apply CGD to the `Qwen2.5-7B-Math` model and compare against multiple strong baselines. We evaluate against: (1) the official Critique Fine-Tuning (CFT) checkpoint [4], trained with GPT-4o as teacher, and (2) our method with both a frontier teacher (Claude Sonnet 3.7) and a weaker open-source teacher (`S1.1-32B`).

The results in Table 3 demonstrate several key findings. First, CGD with Claude Sonnet 3.7 achieves the strongest overall performance (50.4 avg), outperforming CFT (48.9) and representing a **+22.6% gain over the base model** ($27.8 \rightarrow 50.4$). Second, even when using the significantly weaker `S1.1-32B` teacher, CGD maintains competitive performance (49.0 avg, +21.2% over base), demonstrating teacher robustness. This demonstrates again with another model family that CGD can achieve state-of-the-art performance without relying on the most powerful closed-source models, highlighting its practical advantages for resource-constrained settings. Third, the consistent gains across architectures validate that CGD generalizes effectively across model families, scales, and teacher qualities: `LLaMA3.1-8B` (+15.0% on AMC23, +8.0% on Olympiad-Bench), `S1.1-3B` (+10.7% on math reasoning, +3.9% on general reasoning over base model), and `Qwen2.5-Math-7B` (+22.6% over base). Additional ablations on `Mixtral-8x7B` and `OLMo-7B` are provided in Appendix B.

Table 3: **Cross-family validation on `Qwen2.5-Math-7B`.** CGD achieves the strongest performance with both frontier (Claude Sonnet 3.7) and open-source (S1.1-32B) teachers, outperforming CFT (trained with GPT-4o).

| Method | Teacher Model | MATH500 | Minerva-Math | OlympiadBench | AMC23 | AIME24 | Avg. |
|---|---|---|---|---|---|---|---|
| Qwen2.5-Math-7B (Base) | - | 55.4 | 13.6 | 19.9 | 40.0 | 10.0 | 27.8 |
| CFT | GPT-4o | 79.2 | 45.2 | 40.7 | 62.5 | 16.7 | 48.9 |
| **CGD (Ours)** | Claude Sonnet 3.7 | 79.4 | 44.1 | 41.2 | **67.5** | **20.0** | **50.4** |
| **CGD (Ours)** | S1.1-32B | **79.6** | **48.5** | **41.3** | 62.5 | 13.3 | 49.0 |

#### 4.3.1 COMPARISON WITH RL-BASED METHODS

Reinforcement learning (RL) has recently been shown to significantly enhance the reasoning capabilities of LLMs (Shao et al., 2024; DeepSeek-AI et al., 2025). To situate CGD within this line of work, we compare against SimpleRL-Zero (Zeng et al., 2025), an open replication of DeepSeek-R1 framework. We report the official numbers released by the SimpleRL [5] and compare them with our results on both `LLaMA3.1-8B` and `Qwen2.5-7B-Math` base models.

On the `LLaMA3.1-8B` base, CGD provides a more balanced improvement profile than SimpleRL-Zero, surpassing it on key benchmarks like MATH500 (+6.4) and achieving a higher average score. To demonstrate the generalizability of this efficiency, we extend the comparison to the stronger `Qwen2.5-7B-Math` base. Here again, CGD achieves a better average score (50.4 vs. 48.9) while showing significant gains over SimpleRL-Zero on Minerva-Math (+10.6 points, 44.1 vs. 33.5) and OlympiadBench (+3.3 points, 37.9 vs. 41.2).

Table 4: **Comparison with RL-based training (SimpleRL-Zero).** CGD achieves comparable or superior performance to the computationally intensive RL method across two different base models, while requiring 144x less training compute.

| Model | Data Size | GPU Hours | MATH500 | Minerva-Math | OlympiadBench | AMC23 | AIME24 | Avg. |
|---|---|---|---|---|---|---|---|---|
| LLaMA3.1-8B | | | | | | | | |
| + *SimpleRL-Zero* | 8K×12 | 1152 | 23.0 | 9.6 | 5.3 | 15.0 | 0.0 | 10.6 |
| + *CGD* | 50K | 8 | 29.4 | 12.9 | 7.0 | 10.0 | 0.0 | **11.9** |
| Qwen2.5-Math-7B | | | | | | | | |
| + *SimpleRL-Zero* | 8K×12 | 1152 | 77.2 | 33.5 | 37.9 | 62.5 | 33.3 | 48.9 |
| + *CGD* | 50K | 8 | 79.4 | 44.1 | 41.2 | 67.5 | 20.0 | **50.4** |

SimpleRL-Zero requires over 1100 GPU-hours with complex, long-horizon sampling (32×H100 GPUs). In contrast, CGD achieves comparable or superior results with only 8 GPU-hours (8×A100

---

[4] `https://huggingface.co/TIGER-Lab/Qwen2.5-Math-7B-CFT`
[5] `https://github.com/hkust-nlp/simpleRL-reason`

GPUs, substantially weaker than H100s), representing a 144× reduction in training compute. This positions CGD as a cost-effective intermediate training paradigm that can precede reasoning SFT or RL in modern training pipelines. Unlike multi-pass inference methods (e.g., Self-Refine (Madaan et al., 2023)) that incur 3-4× latency increases, CGD adds no inference overhead.

### 4.3.2 THE ROLE OF THE CRITIQUE AS A LEARNING SIGNAL

To isolate the impact of the critique as a learning signal during fine-tuning, we compare our full CGD method against a key ablation variant, *CGD without Critique*. In this ablation, the model is trained on the exact same data and targets, but with the critique removed from the input prompt. This forces the model to learn the transformation from a flawed student answer to the refined answer without explicit guidance.

As shown in Figure 4, the inclusion of the critique during training consistently and significantly improves performance on the challenging reasoning benchmarks. The gains are particularly large on complex reasoning tasks such as Minerva-Math and AMC23. This result demonstrates that the critique is not merely redundant context but is a crucial component of the training signal. It provides an explicit reasoning path that enables the model to learn the difficult self-correction skill more effectively, leading to better generalization on downstream tasks.

**Connection to Diagnostic Findings.** These performance improvements are directly explained by our diagnostic analyses (Appendix C). We find that CGD-trained models exhibit statistically significantly lower entropy ($p < 10^{-4}$) compared to strong baselines when performing self-correction, indicating higher confidence in their reasoning. Moreover, the critique enables 27% more efficient gradient norms during training, providing a clearer optimization signal. Quantitative overlap analysis (Appendix C.6) reveals only 16.6% token overlap and 5.7% bigram overlap between training and test data, confirming that performance gains result from learned reasoning skills rather than memorization. On challenging AIME 2024 problems, CGD achieves 5× higher accuracy than the base model with 4.4× longer reasoning chains (477 vs 2110 words), demonstrating genuine improvement in complex problem-solving ability rather than pattern matching. To understand the mechanisms

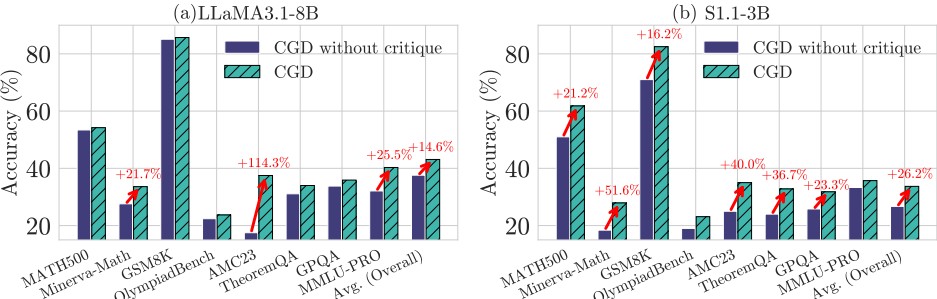

Figure 4: **Performance comparison of CGD with and without the critique as input during training, evaluated on eight benchmarks.** The critique provides a crucial learning signal, leading to consistent accuracy improvements across both the `LLaMA3.1-8B Instruct` (a) and `S1.1-3B` (b) student models.

behind this improvement, we conducted a series of diagnostic probes (see Appendix C for full details and figures). These analyses reveal that the CGD training process creates a more robust and efficient model. We find that the critique provides a more efficient optimization path during training, reducing the required gradient norm by 27%. This efficient learning translates into a model that is statistically significantly more confident (lower entropy, $p < 10^{-4}$) than strong distillation baselines when performing the complex self-correction task. This suggests that CGD's performance gains are driven by its unique ability to instill a robust, decisive, and efficient self-correction capability.

### 4.3.3 TRAINING STABILITY AND HYPERPARAMETER ROBUSTNESS

To ensure a rigorous and fair comparison, we evaluate the learning rate sensitivity of CGD against the CFT baseline using the same teacher model (`LLaMA3.3-70B Instruct`) and identical

prompts from the WebInstruct subset (Table 5). Both methods are trained on 100K critique-augmented examples under identical training schedules, varying only the learning rate between $1 \times 10^{-6}$ and $5 \times 10^{-6}$. While CFT's performance significantly degrades at the higher learning rate, dropping by over 9 points on average, CGD remains robust and outperforms CFT across all metrics regardless of learning rate. These results suggest that CGD's structured self-correction task with the use of both critiques and refined answers enables more stable optimization and better generalization, even under suboptimal hyperparameter choices, whereas CFT remains brittle to training dynamics despite access to the same supervision signals.

Table 5: **Comparison of CGD and CFT using 100K WebInstruct critique-augmented samples.** CGD consistently outperforms CFT across all benchmarks and is relatively robust to learning rate changes, while CFT exhibits significant performance degradation at higher learning rates. The Avg. column reflects average performance across all tasks.

| Method | MATH500 | Minerva-Math | GSM8K | OlympiadBench | AMC23 | TheoremQA | Avg. |
|---|---|---|---|---|---|---|---|
| CFT (LR = $1 \times 10^{-6}$) | 51.8 | 32.7 | 84.8 | 15.7 | 22.5 | 28.5 | 39.3 |
| CFT (LR = $5 \times 10^{-6}$) | 33.4 | 10.3 | 82.9 | 10.1 | 27.5 | 16.1 | 30.1 |
| CGD (LR = $1 \times 10^{-6}$) | 54.2 | 33.6 | 85.7 | 23.7 | 37.5 | 34.0 | 44.8 |
| CGD (LR = $5 \times 10^{-6}$) | 55.0 | 30.1 | 82.3 | 21.6 | 32.5 | 31.9 | 42.2 |

## 5 LIMITATIONS AND FUTURE WORK

While CGD achieves strong results, performance gains depend on the student model's receptivity to critique-conditioned training, influenced by architectural priors and alignment. The multi-stage data generation, though more efficient than RL, incurs upfront computational cost. Preliminary analyses show that CGD serves as a cost-effective intermediate training paradigm that can be used as a warm-start before reasoning SFT or RL, providing a scalable enhancement to modern LLM training workflows.

Notably, CGD improves reasoning without exposing the student to hidden chain-of-thought (CoT) traces, avoiding "thinking+answer" concatenations. Future work could explore integrating CGD with explicit CoT supervision (e.g., combining critiques with intermediate reasoning steps or thinking field as the critique and the answer field as the refined answer), single-stage distillation, and leveraging critiques for safety alignment by penalizing inaccuracies or harmful content.

## 6 CONCLUSION

We introduced CRITIQUE-GUIDED DISTILLATION (CGD), a simple yet powerful fine-tuning framework that teaches models not only what the correct answer is but also why it is correct. By conditioning a student on its own mistake and an explanatory critique, our method learns a robust self-correction skill and preserves answer format without inference-time overhead. Experiments show that CGD significantly outperforms strong baselines across diverse mathematics and general reasoning benchmarks. On `LLaMA3.1-8B` and `S1.1-3B`, this yields average gains of 5.4% and 7.2% over CFT with particularly strong improvements on challenging benchmarks (+15.0% AMC23, +8.0% OlympiadBench). Additional cross-family validation on `Qwen2.5-Math-7B` using both frontier (Claude Sonnet 3.7) and weaker open-source teachers (`S1.1-32B`) achieves state-of-the-art performance (50.4 avg, +22.6% over base) while using 144× less compute than RL methods, confirming robustness across model families, scales, and teacher qualities.

Critically, despite training on data containing no code (WebInstruct spans Math, scientific domains, and Business, but excludes code), CGD generalizes to out-of-distribution benchmarks: +4.88% pass@1 on HumanEval (code generation), and preserved or improved performance on GPQA, MUSR, TruthfulQA, and BBH. By contrast, CFT suffers from catastrophic forgetting with a -21.3% drop on IFEval, highlighting CGD's advantage in preserving general capabilities while improving reasoning. These findings position CGD as a cost-effective intermediate training paradigm that can precede reasoning SFT or RL in modern training pipelines, offering a scalable path toward more capable and reliable language models.

## 7 USE OF LARGE LANGUAGE MODELS (LLMs)

For this submission, large language models (LLMs) were used solely as a general-purpose writing assistant to paraphrase and smooth the authors' original text. LLMs did not generate new scientific content or contribute research ideas. All research questions, methods, analyses, and conclusions were designed and authored entirely by the human researchers. In addition, while LLMs were employed in experiments, their role was strictly as experimental components rather than collaborators in ideation or writing. The authors take full responsibility for all content presented in this paper.

## 8 ETHICS STATEMENT

This work relies exclusively on publicly available or synthetic datasets (e.g., WebInstruct, MetaMathQA). No human subjects, private, or sensitive data were used. The proposed CRITIQUE-GUIDED DISTILLATION (CGD) framework is designed to improve reasoning robustness and efficiency of large language models. We do not anticipate any direct societal risks beyond those already inherent to general LLM research. All authors have read and adhere to the ICLR Code of Ethics.

## 9 REPRODUCIBILITY STATEMENT

We have taken multiple steps to ensure the reproducibility of our results. Section 4 of the main paper details datasets, baselines, and training setups, while Appendix A provides full hyperparameter configurations. Appendix B reports extensive ablations across models, teachers, datasets, and training dynamics. Appendix D includes representative data samples, and Appendix E supplies code instructions with configuration files and scripts. An anonymized code archive is provided in the supplementary materials to enable end-to-end reproduction of our experiments.

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

## SUPPLEMENTARY MATERIAL

In this supplementary material, we provide the following additional details for our work.

- **Appendix A: Experimental Setup Details.** We provide full details on our training and evaluation hyperparameters.
- **Appendix B: Additional Benchmark Results.** We present extensive ablation studies, including results on different student and teacher models, learning rate sensitivity analyses, and training curves.

- **Appendix C: Detailed Diagnostic Analyses.** We provide the full methodology, quantitative results, and qualitative visualizations for the experiments that analyze the internal behavior of the CGD-trained models.

- **Appendix D: Critique and Refinement Generation Prompts.** We provide the exact prompts used for generating teacher critiques and refined answers during training data creation.

- **Appendix E: Data Examples.** We provide a representative training data sample, and qualitative analysis of model responses.

- **Appendix F: Code Instructions.** Provides a summary of the codebase and instructions for reproducing training and evaluation results.

## A EXPERIMENTAL SETUP AND HYPERPARAMETERS

### A.1 EXPERIMENTAL SETUP

All experiments were conducted using NVIDIA A100 40GB GPUs. For training large-scale models, we employed DeepSpeed ZeRO-3 optimization for efficient memory and compute scaling across multiple GPUs, which enables optimizer state partitioning, gradient partitioning, and activation checkpointing to support training with larger batch sizes and model sizes.

We evaluate model performance using exact match accuracy, averaged over the test sets, and report mean performance over three random seeds to account for training variability.

### A.2 HYPERPARAMETERS

We provide the key hyperparameters used in training our models across all experiments. Unless otherwise noted, these values were held constant.

Table 6: **Summary of hyperparameters used in our experiments.**

| Hyperparameter | Value |
|---|---|
| Batch size | 64 |
| Learning rate | 1e-6 |
| Optimizer | AdamW |
| Scheduler type | cosine |
| Max sequence length | 8192 |
| Number of epochs | 1 |
| Warmup ratio | 0.1 |

## B ADDITIONAL EXPERIMENTAL RESULTS

### B.1 ABLATION STUDIES

In this section, we analyze the mechanisms behind CGD's effectiveness and study the impact of ablation studies. We demonstrate CGD's robustness across different training datasets and hyperparameters. We further analyze the impact of the training data's critique composition in Appendix B.1.4, finding that a balanced mixture of feedback yields the most robust model.

### B.1.1 ABLATION: GENERALIZATION TO MATH-SPECIFIC TRAINING DATA

To test the generalizability of our method, we conducted experiments using MetaMathQA, a math-reasoning-focused dataset. As shown in Table 7, CGD again demonstrates strong performance, outperforming all baselines on on Group 1 and Group 2. This confirms that the benefits of the CGD framework are not limited to a specific data source. Notably, CGD surpasses the strongest baseline, i.e., CFT, on the advanced MATH500 and OlympiadBench challenges, yet shows slightly lower

performance on Minerva-Math and GSM8K, which consist of middle-school to undergraduate-level problems.

Table 7: **Comparison of fine-tuning methods on `LLaMA3.1-8B Instruct` across math and reasoning tasks using 100K MetaMathQA examples with LLaMA3.3-70B Instruct as the teacher model.** Optimal results are highlighted in bold, while suboptimal outcomes are underlined. The Avg. columns represent the average performance across Groups 1 and 2, respectively.

| Method | Math Reasoning Tasks (Group 1) | | | | | | General Reasoning Tasks (Group 2) | | | |
|---|---|---|---|---|---|---|---|---|---|---|
| | MATH500 | Minerva-Math | GSM8K | OlympiadBench | AMC23 | Avg. | TheoremQA | GPQA | MMLU-PRO | Avg. |
| LLaMA3.1-8B Instruct | 50.6 | 33.5 | 85.3 | 14.5 | 22.5 | 41.3 | 27.6 | 30.8 | 31.2 | 29.9 |
| + *SFT* | 47.8 | 29.8 | 85.5 | 13.6 | 27.5 | 40.8 | 28.1 | **32.8** | 37.0 | 32.6 |
| + *Distilled SFT* | 50.2 | 33.5 | 79.8 | 18.5 | **35.0** | 43.4 | 31.2 | 28.8 | 28.1 | 29.4 |
| + *CFT* | 52.8 | **36.4** | **88.6** | 17.2 | 32.5 | 45.5 | 31.1 | 30.7 | **38.3** | 33.4 |
| + *CGD* | **59.0** | 34.6 | 87.3 | **21.8** | 32.5 | **47.0** | **34.1** | 30.3 | 36.1 | **33.5** |
| Δ = CGD - CFT | 6.2 | -2.2 | -1.3 | 4.6 | 0.0 | 1.5 | 3.0 | -0.5 | -2.2 | 0.1 |

## B.1.2 ABLATION: RESULTS ON DIFFERENT STUDENT MODELS

To assess the generality of our approach beyond the LLaMA model family, we replicate our main fine-tuning comparisons using `Mixtral-8x7B Instruct v0.1` and `OLMo-2-1124-7B-Instruct` as the student models. Table 8 summarizes the results for both `Mixtral-8x7B Instruct v0.1` and `OLMo-2-1124-7B-Instruct` student models across both math-focused and general reasoning benchmarks, with all models trained on the same 100K WebInstruct prompts.

Table 8: **Evaluation of fine-tuning methods on Mixtral-8x7B Instruct across math-focused (Group 1) and general reasoning (Group 2) benchmarks, using WebInstruct as the training set.** CGD achieves the strongest performance in both groups, despite Mixtral being a different architecture than LLaMA. All methods are fine-tuned on 100K WebInstruct samples. Bold numbers denote the best, and underlined values indicate the second-best performance. The Δ row shows CGD's gains over the CFT baseline.

| Method | Math Reasoning Tasks (Group 1) | | | | | | General Reasoning Tasks (Group 2) | | | |
|---|---|---|---|---|---|---|---|---|---|---|
| | MATH500 | Minerva-Math | GSM8K | OlympiadBench | AMC23 | Avg. | TheoremQA | GPQA | MMLU-PRO | Avg. |
| Mixtral-8x7B Instruct | 29.6 | 15.4 | 69.4 | 8.9 | 7.5 | 26.2 | 21.2 | 21.7 | 24.7 | 22.5 |
| + *SFT* | 31.4 | 15.6 | 65.6 | 7.9 | 5.0 | 25.1 | 20.4 | 20.5 | **25.2** | 22.0 |
| + *CFT* | 35.6 | 20.6 | 63.8 | 11.1 | **10.0** | 28.2 | 23.6 | **31.8** | 16.0 | 23.8 |
| + *CGD* | **39.0** | **23.9** | **75.0** | **11.7** | 7.5 | **31.4** | **26.4** | 25.8 | 23.3 | **25.1** |
| Δ = CGD - CFT | 3.4 | 3.3 | 11.2 | 0.8 | -2.5 | 3.2 | 2.8 | -6.0 | 7.3 | 1.3 |
| OLMo-2-1124-7B-Instruct | 35.4 | 16.5 | 81.9 | 11.0 | 7.5 | 30.5 | 23.0 | 28.3 | 34.1 | **28.5** |
| + *SFT* | 36.4 | 15.1 | 80.5 | 11.0 | 12.5 | 31.1 | 19.1 | 28.1 | 34.1 | 27.2 |
| + *CFT* | 35.9 | 16.8 | 81.2 | 11.8 | 10.0 | 31.1 | 19.3 | 27.4 | 33.4 | 26.7 |
| + *CGD* | **37.4** | **16.9** | **83.2** | **12.1** | **20.0** | **33.9** | **24.2** | **28.3** | **34.2** | 28.2 |
| Δ = CGD - CFT | 1.5 | 0.1 | 2.0 | 0.3 | 10.0 | 2.8 | 4.9 | -1.1 | 0.7 | 1.5 |

Notably, we find that our method, CGD, consistently outperforms the baselines in both task groups for a different student model `Mixtral-8x7B Instruct` as shown in Table 8. On math reasoning tasks (Group 1), CGD achieves a +3.2% improvement over CFT. This includes substantial gains on GSM8K (+11.2%), Minerva-Math (+3.3%), and MATH500 (+3.4%), confirming transferability to a different architecture. In general reasoning tasks (Group 2), CGD shows a +1.3% average improvement over CFT, with notable gains on MMLU-PRO (+7.3%) and TheoremQA (+2.8%). While performance slightly declines on AMC23 (-2.5%) and GPQA (-6.0%) relative to CFT, these drops are not large enough to offset the overall performance improvements.

In contrast, CGD yields smaller gains on OLMo, i.e., 0.4 points less gain on Group 1 Avg. compared to Mixtral. While OLMo and Mixtral are similar in scale and baseline strength, they may differ in their ability to absorb critique-structured inputs. One possible explanation is differences in alignment data quality and fine-tuning objectives: prior work ( (Bai et al., 2022; Liang et al., 2025; Moon et al., 2025; Liu et al., 2024)) suggests that models tuned with richer dialogue-style data better leverage multi-step feedback. These results highlight that CGD is most effective when the student has been trained with supervision formats resembling critique/refinement, and they motivate deeper investigation into model-specific receptivity to critique-based training.

Importantly, CGD achieves consistently higher scores than SFT and CFT across most benchmarks, suggesting that distillation from critiques offers a more stable supervision signal than critique generation alone. These results generalize our main findings and further support the modularity and versatility of our proposed training framework, highlighting that critique-based supervision is effective even for non-LLaMA models.

**Understanding Variation in Gains Across Model Families.** Our experiments reveal an important empirical pattern: CGD shows consistent improvements across model families, but the magnitude varies. Based on existing literature on instruction-tuning and data quality (Bai et al., 2022; Liang et al., 2025; Moon et al., 2025; Liu et al., 2024), we believe two factors best account for this variation: **(A) alignment data quality and receptivity to critique-structured inputs**, and **(B) architectural and pretraining-induced inductive biases**.

Prior work has shown that the quality and format of instruction-tuning data strongly affects downstream alignment and task performance (Bai et al., 2022; Liang et al., 2025). Models exposed to richer, more diverse, or interactive alignment data tend to make better use of supervision signals structured as dialogue, critique, or contrastive feedback. This is consistent with our empirical pattern:

- **LLaMA3.1-8B** (extensively instruction-tuned): large gains (+15.0% AMC23, +8.0% Olympiad-Bench, +5.4% math avg over CFT)

- **S1.1-3B** (math-specialized with strong reasoning priors): large gains (+10.7% math reasoning avg over base, +7.2% over CFT)

- **Qwen2.5-Math-7B** (math-specialized with strong reasoning priors): large gains (+22.6% over base, 27.8% $\rightarrow$ 50.4% avg)

- **Mixtral-8x7B** (MoE): modest gains (+3.2% avg over CFT)

- **OLMo-7B** (different pretraining corpus): modest gains (+2.8% avg over CFT)

These correlations suggest that a model's prior alignment to supervision and its baseline reasoning ability can amplify the benefit of CGD. While our current results support this interpretation, future work needs to perform more controlled experiments to gain deeper insights into this important phenomenon.

### B.1.3 ABLATION: RESULTS USING DIFFERENT TEACHER MODELS

We find that CGD provides consistent improvements over the base `LLaMA3.1-8B Instruct` model across both math and general reasoning benchmarks, regardless of the choice of teacher model as shown in Table 9. Using `LLaMA3.3-70B Instruct` as the teacher yields strong gains, particularly in general reasoning tasks, while adopting the open-weight S1.1-32B teacher leads to even stronger performance on several challenging math benchmarks. For example, CGD with S1.1 improves AMC23 accuracy by +20.0 absolute points (22.5 $\Rightarrow$ 42.5). These results suggest that the benefits of CGD are not limited to teacher scale or architecture family; even when transferring critiques from a non-LLaMA teacher, the student acquires improved reasoning ability.

We emphasize that the teacher ablation in Table 9 holds the student fixed (`LLaMA3.1-8B`) while varying the teacher model. In contrast, the `S1.1-3B` results presented in Table 1 focus on the student-side generalization, where the model itself is smaller and trained with critiques and responses from `S1.1-32B`.

Importantly, these findings support the claim that CGD's effectiveness is not solely determined by the raw strength of the teacher, but also by the structured way in which critiques are generated and incorporated during training. While stronger teachers such as GPT-4o or future generations of S1.1 may offer further improvements, our preliminary experiments already demonstrate that critique quality and integration play a critical role in driving gains. In other words, CGD does more than transfer answers, i.e., it teaches the student how to reason through structured critique, enabling performance improvements that extend beyond what is achievable with standard distillation.

Table 9: **Comparison of CGD using different teacher models on the student model `LLaMA3.1-8B Instruct` across math (Group 1) and general reasoning (Group 2) benchmarks, using WebInstruct as the training set.** Using `S1.1` as the teacher model achieves a stronger performance in complex math-reasoning tasks, despite `S1.1` being a different architecture than LLaMA.

| Method | Math Reasoning Tasks (Group 1) | | | | | | General Reasoning Tasks (Group 2) | | | |
|---|---|---|---|---|---|---|---|---|---|---|
| | MATH500 | Minerva-Math | GSM8K | OlympiadBench | AMC23 | Avg. | TheoremQA | GPQA | MMLU-PRO | Avg. |
| Initialized from LLaMA3.1-8B Instruct | | | | | | | | | | |
| LLaMA3.1-8B Instruct | 50.6 | 33.5 | 85.3 | 14.5 | 22.5 | 41.3 | 27.6 | 30.8 | 31.2 | 29.9 |
| *+ CGD with LLaMA3.3-70B* | 54.2 | 33.6 | 85.7 | 23.7 | 37.5 | 46.9 | 34.0 | 35.9 | 40.3 | 36.7 |
| *+ CGD with S1.1-32B* | 56.8 | 37.1 | 86.8 | 16.7 | 42.5 | 48.0 | 32.2 | 34.3 | 40.4 | 35.7 |
| Teacher Models | | | | | | | | | | |
| LLaMA3.3-70B Instruct | 75.3 | 55.9 | 96.1 | 39.3 | 65.0 | 66.3 | 53.6 | 37.9 | 70.6 | 54.0 |
| S1.1-32B | 92.9 | 58.1 | 94.8 | 63.6 | 85.0 | 78.9 | 64.4 | 46.0 | 48.3 | 52.9 |

### B.1.4 ABLATION: IMPACT OF CRITIQUE CORRECTNESS MIXTURE

To investigate the impact of the training data composition, we conducted an ablation study using the WebInstruct dataset, training five models on data with varying ratios of correct and incorrect student answers (as indicated by the critique's conclusion). We kept the total sample size (25k) and all other hyperparameters identical across runs. The results averaged over our math reasoning benchmarks (MATH500, Minerva-Math, etc.), are shown in Figure 5. We observe a non-linear relationship: models trained on a balanced mixture of both correct and incorrect examples (specifically the 50/50 split) achieve the highest performance. This suggests that for a model to learn a truly generalizable self-correction skill, it must be exposed to a diverse range of both positive and negative feedback, preventing it from learning a simple heuristic like "always agree with the critique".

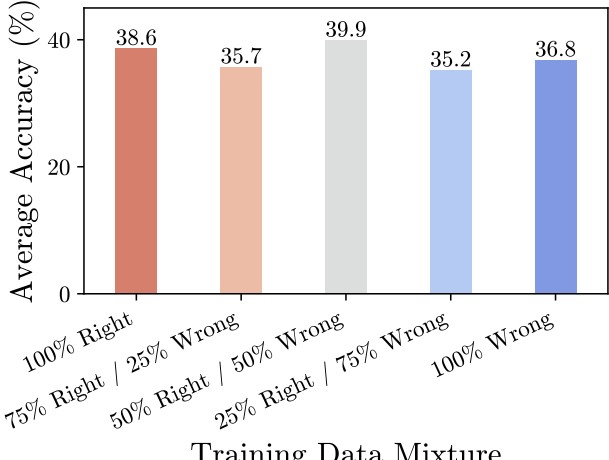

Figure 5: **Performance on an average of math benchmarks for models trained on different mixtures of correct/incorrect student answers.** A balanced 50/50 mixture yields the most robust model.

### B.2 EPOCH-ACCURACY CURVES

Figure 6 shows the progression of final accuracy across training epochs for CRITIQUE-GUIDED DISTILLATION (CGD) on six math-focused benchmarks. We observe that performance is generally stable throughout training, with no substantial drops in accuracy for any dataset. While the upward trends are not particularly pronounced, the lack of degradation suggests that our method is robust to overfitting and avoids catastrophic forgetting. In particular, benchmarks such as MATH (increases from 55.8 to 56.7) and OlympiadBench (increases from 22 to 23.3) show modest improvements, indicating some continued learning over time. These curves offer cautious empirical support for the consistency and stability of our fine-tuning process.

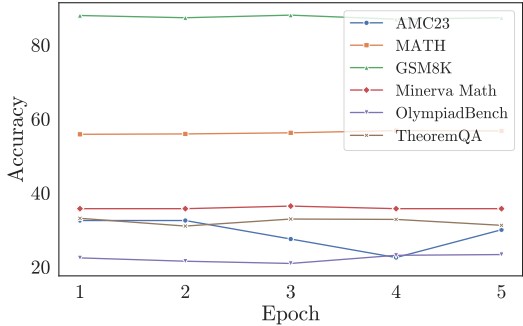

Figure 6: **Accuracy over training epochs for CGD on six math-focused benchmarks.** While trends are modest, performance remains stable throughout, indicating resistance to overfitting and catastrophic forgetting.

### B.3 LEARNING-RATE SENSITIVITY

Figure 7 depicts how both methods respond to changes in learning rate. Figures (a) and (b) show the accuracy vs. learning-rate curves for our approach and CFT, respectively. Our method exhibits a smooth decline as the learning rate increases (Fig. 7a), whereas CFT's performance degrades more sharply (Fig. 7b).

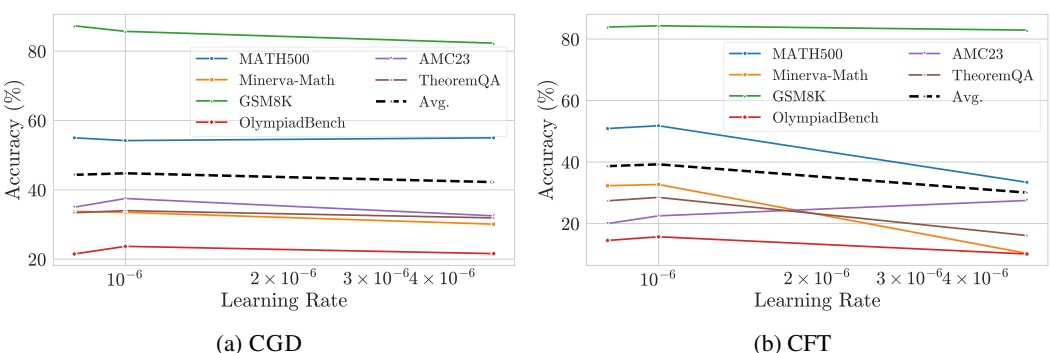

Figure 7: **Accuracy vs. learning rate for (a) CGD (our method) and (b) the CFT baseline across six benchmarks.**

### B.4 TRAINING LOSS ANALYSIS

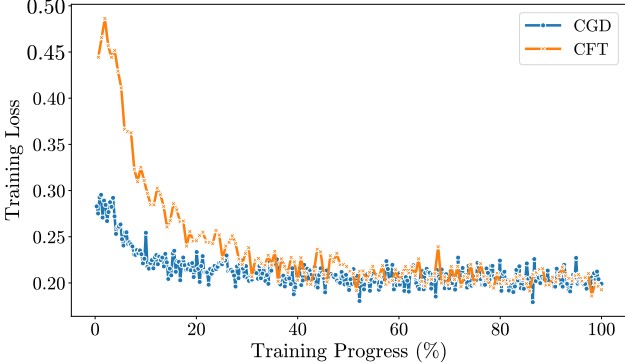

Figure 8: **Training loss comparison between CGD and CFT.** The x-axis indicates normalized training progress, and the y-axis shows loss.

We present training loss curves comparing CRITIQUE-GUIDED DISTILLATION (CGD) and Critique-Finetuning (CFT) methods in Figure 8. The x-axis represents normalized training progress (%), and the y-axis shows the training loss.

From the plot, the CFT curve exhibits a noticeable initial spike in loss, which can be attributed to a format drift during early training. This is due to the model being trained on a critique-style instruction following dataset immediately after pretraining or SFT tuned on QA-style instructions. The shift from generating answers to critiquing Q&A pairs likely introduces a mismatch in expected input-output format, temporarily destabilizing the loss. As training progresses, however, the model adapts, and the loss curve stabilizes and declines.

In contrast, the CGD method shows a more stable and smooth decrease in loss throughout training, suggesting a more consistent and format-aligned supervision signal. This supports the hypothesis that CGD, by leveraging structured critiques without drastic task shifts, offers a gentler optimization trajectory and better alignment with initial model capabilities.

## C    DETAILED DIAGNOSTIC ANALYSES

This section provides the detailed methodology, full quantitative results, and visual analyses for the diagnostic experiments summarized in the main paper. All diagnostic experiments were conducted on a set of 500 samples randomly drawn from the OpenMathInstruct 2 dataset using `LLaMA3.1-8B Instruct` student model with the same hyperparameters.

### C.1    DIAGNOSTIC EXPERIMENTS

**Entropy Calculation.**    To measure predictive confidence, we performed a forward pass for each model on the diagnostic dataset, using the full '(Prompt + Student Answer + Critique)' context. This context is formatted using the model's specific chat template. We then isolated the model's logits for the single, next token that would begin the 'Refined Answer'. These logits were converted to a probability distribution via the softmax function, and the Shannon entropy $(H(X) = -\sum p(x) \log p(x))$ was calculated. A lower entropy value indicates higher confidence in the prediction.

**Gradient Norm Calculation.**    To measure learning signal efficiency, we took each final trained model and performed a single forward and backward pass on a diagnostic sample to compute the cross-entropy loss against the target answer. We then calculated the total L2 norm of the full parameter gradient vector. This was done for two input conditions: one 'With Critique' and one 'Without Critique', allowing for a controlled analysis of the critique's impact on the update signal. Analysis of Table 11 reinforces the fact that conditioned on an informative critique the model is able to better predict the final response. This reduces the loss and in turn the magnitude of the gradient norm. We observed this trend during the entire period as well.

**Attention Analysis.**    To analyze the model's internal reasoning, we generated answers with maximum 8192 tokens for each sample and collected the attention matrices from all 32 layers. These scores were then aggregated by averaging across all attention heads and normalized to represent the percentage of attention paid by each generated token to three distinct sections of the prompt: the 'Problem', the 'Student Answer', and the 'Critique'.

### C.2    QUANTITATIVE ANALYSIS OF MODEL CONFIDENCE

The behavioral differences observed in our case study are supported by our quantitative diagnostics. As shown in Table 10, the key finding is that the CGD model is statistically significantly more confident (lower entropy) than all other generative baselines on the self-correction task. The statistical significance of this result ($p < 10^{-4}$ vs. Distilled SFT) confirms that the CGD training process forges a uniquely robust and decisive reasoning agent.

### C.3    ATTENTION MECHANISM ANALYSIS

To provide a deeper mechanistic view of the CGD model's reasoning process, we analyzed its internal attention patterns, averaged over 50 samples from the OpenMathInstruct 2 dataset. For each

Table 10: **Summary of predictive confidence (Mean Entropy), averaged over 500 samples from OpenMathInstruct 2.** Lower entropy is better. Significance markers (*, **, ***) denote the p-value of a paired t-test comparing each baseline to our CGD model.

| Model | Mean Entropy |
|---|---|
| Baseline SFT | 6.56*** |
| SFT | 6.62*** |
| Distilled SFT | 6.49*** |
| **CGD** | 6.44 |

*Significance: \*\*\* $p < 0.001$*

Table 11: **Gradient norm analysis for the final trained CGD model.** The presence of a critique provides a more efficient signal, reducing the update magnitude by 27%.

| CGD Model Condition | Mean Gradient Norm | Std. Dev. Gradient Norm |
|---|---|---|
| Without Critique | 2446.9 | 2011.9 |
| With Critique | 1802.7 | 1765.5 |

sample, we generated up to 8192 new tokens, allowing the model to complete its reasoning naturally. Our key finding is that the model employs a sophisticated, multi-phase reasoning strategy, using the critique as a foundational signal that is internalized early and acted upon during generation. This is illustrated across three complementary visualizations.

Figure 9 presents the model's attention flow across different layers during the generation of an answer. Starting at the first layer and all the way to the middle layers, there is significant attention on both the critique and the student response. This shows that the model has learned to exploit the signals in an informative critique and the noisy student response (e.g., with attention to the **Critique** at 48.1% and the **Student Answer** at 36.0% at the very first generation step). In the later layers of the model, the primary focus is on getting the correct response and hence most of the attention is on the problem.

Figure 10 confirms how different pieces of information are processed at different levels of abstraction. The plot shows the average attention paid to each prompt section across all 32 layers. The results show that direct attention to the **Critique**'s raw tokens peaks at the very input (31.9% at Layer 0), suggesting a strong initial intake of the signal. The model's focus then shifts to the **Student Answer**, with attention peaking in the semantic middle layers (22.1% at Layer 13), precisely where attention to the critique also sees a secondary rise. This could be attributed to the fact that the model's most abstract reasoning, understanding the flaw and synthesizing the correction, happens in the middle of the network. Finally, attention to the **Problem** details consolidates and peaks in the late layers (94.8% at Layer 25) as the model formulates its final output.

Finally, Figure 11 provides a high-level summary of attention from different layers, broken down by generation phase, which reinforces these findings. The heatmaps for later, more semantic layers (16 and 31) visualize the "plan-then-execute" pattern, showing that the initial generation phases are dominated by attention to the critique (48.1% for Layer 31 at token 1). This is consistent with a model that has learned to use the critique as a foundational guide to initiate and structure its reasoning process. These observed attention patterns suggest that the CGD has acquired a sophisticated reasoning process: it internalizes the critique's guidance at an early stage and then acts upon this internalized knowledge in its final, semantic layers to plan and execute a corrected solution. The following section provides a direct behavioral test of this hypothesis.

## C.4 CASE STUDY: COUNTERFACTUAL ANALYSIS

To test whether CGD learns a functional skill of robust reasoning beyond simple contextual understanding, we performed a qualitative case study. We presented both the baseline Llama 3.1 Instruct model and our final CGD-trained model with a problem from our test set under two conditions. In the **Factual** condition, we provided the original, correct critique from our dataset. In the **Counter-**

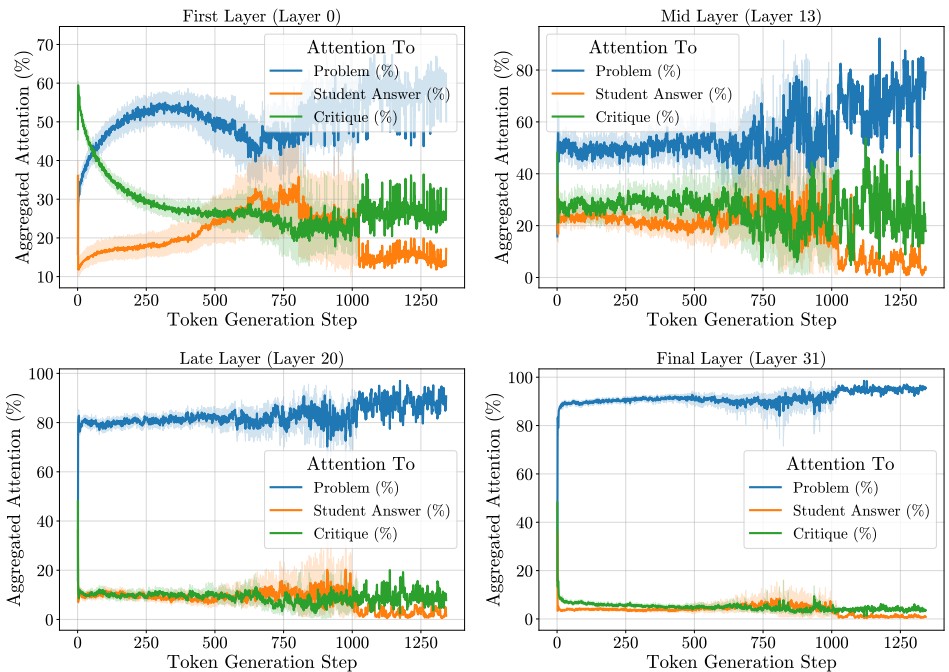

Figure 9: **Average attention flow of the CGD model.** All layers shown begin with a "planning" step, focusing on the Critique (48.1%) and Student Answer (36.0%). The final layer (bottom right) then pivots sharply to an "execution" phase, focusing on the Problem ($> 90\%$), while the first layer (top left) continues to process the Critique. Shaded regions represent the 95% confidence interval over 50 samples.

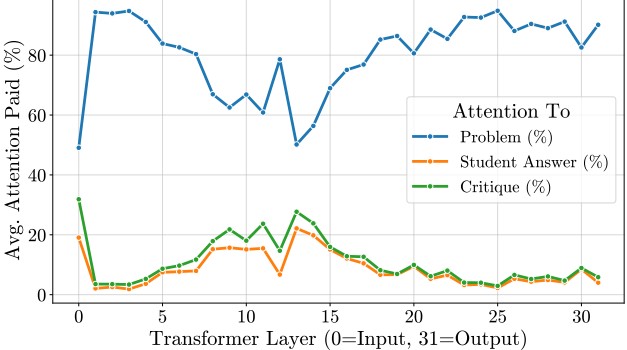

Figure 10: **Average attention paid to each prompt section across all 32 transformer layers.** The patterns suggest an *early intake* of the Critique (peak at Layer 0), followed by a *deep processing* of the Student Answer in conjunction with the critique in the semantic middle layers (peak at Layer 13). Attention to the **Problem** dominates in the final layers.

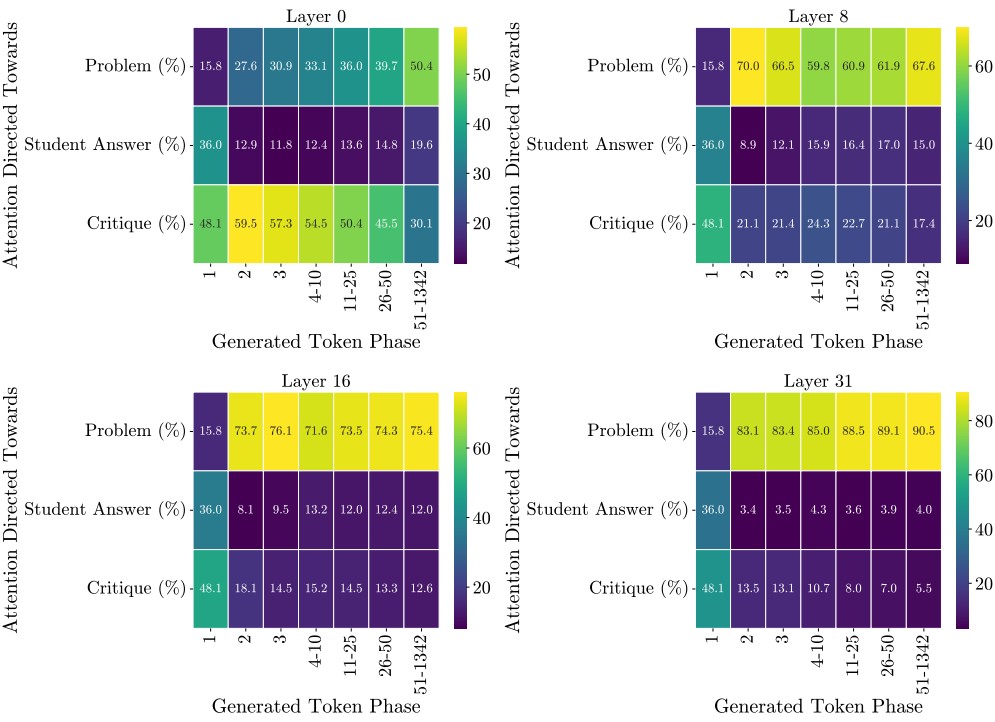

Figure 11: **Aggregated heatmap of attention by generation phase for representative layers.** The bright cells for the **Critique** in the first column for Layer 16 and 31 (48.1% and 45.5%) confirm that the initial planning phase is critique-driven, acting on the signal internalized by the early layers. The sustained brightness for the Critique in the Layer 0 heatmap illustrates its role in early-stage processing.

**factual** condition, we provided a generic but nonsensical critique that was irrelevant to the problem (e.g., incorrectly referencing a "quadratic formula" for a number theory problem). We then evaluated the correctness of the final generated answer in all four scenarios. Table 12 summarizes the results for a representative sample.

The results demonstrate a clear behavioral divergence. While both models produce the correct answer when given a factual critique, their responses to flawed guidance differ significantly. The baseline model proves brittle; it is functionally *derailed* by the nonsensical critique, failing to produce a correct answer. The CGD model, in contrast, exhibits superior robustness by **ignoring the misleading critique** and producing the correct answer based on its own problem analysis.

**Interpretation.** This counterfactual analysis provides critical evidence that CGD models learn reasoning principles rather than critique obedience. The model's ability to ignore misleading critiques demonstrates it internalized: (1) mathematical reasoning rules, (2) critique quality assessment, and (3) independent problem-solving capability. This robustness emerges from exposure to diverse critique qualities during training—recall that our data mixture ablation (Figure 5 in main paper) shows performance peaks with 50/50 correct/incorrect answer mixture rather than corrections-only. The model learns to evaluate critique validity against its own understanding rather than blindly trusting external feedback, developing an internal validation mechanism that distinguishes informative from non-informative guidance.

Table 12: **Counterfactual analysis results for a representative sample (Sample #42).** While both models are correct with factual guidance, only the CGD model is robust to the flawed, counterfactual critique.

| Model | Outcome (Factual Critique) | Outcome (Counterfactual Critique) |
|---|---|---|
| LLaMA3.1-8B Instruct | CORRECT | INCORRECT (Derailed) |
| CGD | CORRECT | CORRECT (Robust) |

---

**Case Study: Counterfactual Analysis on Sample #42**

The goal of this analysis is to test for a functional understanding of critiques. The problem's gold-standard answer is **2310**.

---

BASELINE MODEL ANALYSIS

**Condition: Factual Critique**
**Correctness: CORRECT**
**Full Output:**

```
To find the smallest positive integer divisible by five distinct prime
↪  numbers, we need to understand the concept of prime numbers
↪  first...
...
2 * 3 * 5 * 7 * 11 = 2310
Therefore, the smallest positive integer divisible by five distinct
↪  prime numbers is 2310. This result is a direct consequence of the
↪  multiplication of the smallest prime numbers, which are the
↪  fundamental building blocks of all other numbers.
```

**Condition: Counterfactual Critique**
**Correctness: INCORRECT** (Derailed by the flawed critique.)
**Full Output:**

```
...A more accurate approach would be to understand that the smallest
↪  positive integer... is actually their least common multiple
↪  (LCM)...
To find the LCM of 2, 3, 5, 7, and 11, we can list the multiples of
↪  each number and find the smallest number that appears in all the
↪  lists:

Multiples of 2: 2, 4, 6, 10, 14, 22, 30, 42, 70, 110, 154, 330,...
```

```
Multiples of 3: 3, 6, 9, 12, 15, 18, 21, 24, 27, 30, 33, 36, 39, 42,
↪  45, 48, 51,..., 1542, 1545, 1548, 1551, 1554, 1557, 1560, 1563,
↪  1566, 1569,
```

---

CGD-TRAINED MODEL ANALYSIS

**Condition: Factual Critique**
**Correctness: CORRECT**
**Parsed Answer:** 2310
**Full Output:**

```
## Finding the Smallest Positive Integer Divisible by Five Distinct
↪  Prime Numbers

To find the smallest positive integer divisible by five distinct prime
↪  numbers, we need to follow a straightforward approach that
↪  involves identifying the five smallest distinct prime numbers and
↪  then multiplying them together. This method is based on the
↪  fundamental principle that prime numbers are the building blocks
↪  of all other numbers, and ...

### Definition of Prime Numbers

Before proceeding, it's essential to understand what prime numbers
↪  are. Prime numbers are natural numbers greater ...

### Identifying the Five Smallest Distinct Prime Numbers

The first five prime numbers in ascending order are 2, 3, 5, 7, and
↪  11. These numbers are distinct, meaning they are all different
↪  from one another, ...

### Multiplication of the Prime Numbers

To find the smallest positive integer divisible by these five distinct
↪  prime numbers, we multiply them together:

2 * 3 * 5 * 7 * 11 = 2310

This calculation is straightforward and demonstrates the direct
↪  application of the initial logic. By multiplying the smallest
↪  prime numbers, ...

### Conclusion

Therefore, the smallest positive integer divisible by five distinct
↪  prime numbers is indeed 2310. This result is a direct consequence
↪  of multiplying the five smallest distinct prime numbers together.
↪  The approach taken is logical and mathematically sound, ...

In summary, the process of finding the smallest positive integer
↪  divisible by five distinct prime numbers involves identifying the
↪  five smallest prime numbers, which are 2, 3, 5, 7, and 11, and
↪  then multiplying them together. This method ensures that the
↪  resulting product is the smallest possible integer that is
↪  divisible by each of the five distinct primes, making 2310 the
↪  correct answer to the problem.
```

**Condition: Counterfactual Critique**
**Correctness: CORRECT**
**Parsed Answer:** 2310
**Full Output:**

```
To find the smallest positive integer divisible by five distinct prime
↪   numbers, we need to follow a systematic approach that involves ...
↪   However, the initial step involves understanding what prime
↪   numbers are and how to identify them.

Prime numbers are natural numbers greater than 1 that have no positive
↪   ...

The five smallest distinct prime numbers are indeed 2, 3, 5, 7, and
↪   11. These are the first five prime numbers in the sequence of
↪   prime numbers.

Now, to find the smallest positive integer divisible by these five
↪   distinct prime numbers, ...

The calculation is as follows:
2 * 3 * 5 * 7 * 11 = 2310

Therefore, the smallest positive integer divisible by five distinct
↪   prime numbers is indeed 2310. This number is the product of the
↪   five smallest distinct prime numbers, ensuring that it is
↪   divisible by each of them without leaving a remainder.

In conclusion, the approach to finding the smallest positive integer
↪   divisible by five distinct prime numbers involves identifying the
↪   prime numbers, understanding their properties, and then
↪   calculating their product. The result, 2310, is the smallest
↪   number that meets the criteria of being divisible by five distinct
↪   prime numbers.
```

**Conclusion:** The CGD model exhibits superior robustness and instruction-following. It correctly solves the problem under both conditions while adhering to the specified output format. The baseline model, while capable of reasoning correctly, is brittle to both complex instructions and irrelevant, noisy feedback.

## C.5 BAYESIAN INTERPRETATION

Finally, we interpret critique conditioning as a Bayesian posterior update. Let the student's initial output $y'$ define a prior distribution $p(y|x)$, and let the critique $c$ provide new evidence about correctness. The teacher's refinement can be viewed as a posterior distribution:

$$\underbrace{S_\theta(\hat{y} \mid x, y', c)}_{\text{Student posterior}} \propto \underbrace{T_\phi(c \mid x, y', \hat{y})}_{\text{Teacher likelihood}} \times \underbrace{S_{\text{init}}(\hat{y} \mid x, y')}_{\text{Student prior}}. \qquad (2)$$

Here, $S_{\text{init}}(\hat{y} \mid x, y')$ is the student's original (prior) distribution over responses, while $T_\phi(c \mid x, y', \hat{y})$ acts as a scoring function that up-weights those $\hat{y}$ values better aligned with the critique. Note that the teacher "likelihood" need not be normalized; the proportionality sign indicates that normalization is implicit when forming the posterior.

In practice, CGD minimizes the KL divergence between the student's posterior and the teacher-defined target distribution, which directly implements Equation 2 in training via Algorithm 1. This interpretation highlights how critique guidance sharpens the student's prior into a more informative posterior, explaining the observed empirical gains.

## C.6 CRITIQUE-ANSWER OVERLAP ANALYSIS

To directly address concerns about potential answer leakage from critiques, we conducted token-level and phrase-level overlap analysis on 50,000 training examples from our WebInstruct dataset.

**Methodology.** We computed two metrics: (1) **Token-level overlap**: percentage of unique tokens appearing in both critique and refined answer, and (2) **Bigram-level overlap**: percentage of con-

secutive two-word sequences (bigrams) shared between critique and refined answer. Token overlap measures vocabulary sharing, while bigram overlap detects phrase-level copying.

**Results.** Our analysis reveals:

- **Token overlap: 16.6%** — Critiques and answers share individual mathematical terms (e.g., variables like $x$, $y$; operations like "differentiate", "solve"; concepts like "equation", "derivative"). These tokens are universal mathematical vocabulary appearing in the problems themselves.

- **Bigram overlap: 5.7%** — Despite sharing vocabulary, critiques and answers combine terms into different reasoning chains. Only 5.7% of consecutive word pairs overlap.

**Interpretation.** The low bigram overlap (5.7%) despite moderate token overlap (16.6%) demonstrates that the model learns **concepts** (which mathematical terms to use) without memorizing **patterns** (how to phrase solutions). This is learning "how to reason" rather than "what to copy." The model internalizes three content-independent capabilities: (1) mathematical vocabulary, (2) concept application (when to use which operations), and (3) reasoning structure (how to organize multi-step solutions). These capabilities transfer to new problems regardless of critique presence at inference.

### C.7 REASONING QUALITY ON HARD PROBLEMS (AIME 2024)

To provide direct evidence that CGD develops genuine reasoning capability rather than pattern memorization, we analyzed reasoning quality on AIME 2024, a set of 30 extremely challenging competition mathematics problems.

**Methodology.** We compared CGD-trained `LLaMA3.1-8B` against the base model on identical test problems, measuring: (1) accuracy (Pass@1), (2) average reasoning length (words), (3) average reasoning steps, and (4) number of problems solved exclusively by each method.

Table 13: **Reasoning quality analysis on AIME 2024.** CGD generates significantly more detailed reasoning.

| Metric | Base Model | CGD | Improvement |
|---|---|---|---|
| Accuracy | 3.3% (1/30) | 16.7% (5/30) | +13.3% (5×) |
| Avg reasoning length | 477 words | 2110 words | 4.4× longer |
| Avg reasoning steps | 16.4 | 49.5 | 3.0× more |
| Problems only this method solves | 0 | 4 | +4 |

**Results.**

**Interpretation.** CGD achieves 5× higher accuracy and generates 4.4× more detailed step-by-step reasoning on identical hard problems. The dramatic increase in reasoning detail ($477 \rightarrow 2110$ words) demonstrates that CGD learns to produce comprehensive mathematical explanations, not just answer patterns. This provides direct evidence that the model doesn't rely on memorized patterns—it generates vastly more detailed reasoning than its base model on problems it has never seen, proving it learned "how to reason" through critiques. The 4.4× increase in reasoning length and 3.0× increase in reasoning steps show the model internalized mathematical problem-solving strategies that transfer to new, difficult problems.

## D CRITIQUE AND REFINEMENT GENERATION PROMPTS

For transparency and reproducibility, we provide the exact prompts used to generate critiques and refined answers from the teacher model during the CGD training data creation process.

## D.1 CRITIQUE GENERATION PROMPT

> **Teacher Prompt for Generating Critiques**
>
> You are an expert in mathematics and reasoning. Your task is to carefully review a student's solution to a given problem and provide a detailed, constructive critique.
> **Problem:** {problem}
> **Student's Solution:** {student_answer}
> Please analyze the student's solution and provide a critique that:
>
> - Identifies any errors, misconceptions, or gaps in reasoning
> - Explains *why* these issues are problematic
> - Suggests the correct approach or concepts needed
> - Is clear, specific, and pedagogically helpful
>
> Your critique should focus on the reasoning process and help guide the student toward the correct solution without directly providing the full answer.

## D.2 REFINED ANSWER GENERATION PROMPT

> **Teacher Prompt for Generating Refined Answers**
>
> You are an expert in mathematics and reasoning. Given a problem, a student's initial attempt, and a critique of that attempt, provide a complete and correct solution that addresses all the issues identified in the critique.
> **Problem:** {problem}
> **Student's Initial Solution:** {student_answer}
> **Critique:** {critique}
> Please provide a refined, complete solution that:
>
> - Addresses all issues identified in the critique
> - Shows clear, step-by-step reasoning
> - Arrives at the correct answer
> - Maintains mathematical rigor and clarity

## E  EXAMPLE CGD TRAINING DATA SAMPLE

To better understand how CRITIQUE-GUIDED DISTILLATION (CGD) enables improved response quality without format drift, we present a representative CGD training data sample. The CGD training pipeline involves three components: a student's initial answer, a teacher-generated critique that identifies specific flaws and explains the reasoning behind them, and a refined answer that incorporates the critique to correct the initial output. The full input tuple (prompt, student, critique, refined answer) is only used during training, and inference remains unchanged from standard decoding.

In the example, the student incorrectly assumes the number of electrons per nitrogen atom when calculating the mass of a nitride anion. The critique explicitly identifies this conceptual error, and the refined answer provides the correct computation with an explanation of the correct electronic configuration of the ion. This structured format teaches the model both the nature of the mistake and the reasoning needed to arrive at a correct solution, demonstrating how CGD effectively enhances model performance while preserving answer format consistency.

> **CGD WebInstruct Training Data Sample**
>
> **Question:** Calculate the mass of a nitride anion that contains $2.408 \times 10^{24}$ nitrogen-based electrons.
> **Student's Initial Answer:** To calculate the mass of the nitride anion, we first need to determine the number of nitrogen atoms in the anion. Since each nitrogen atom has 7 electrons, we can divide the total number of electrons by 7 to find the number of nitrogen atoms:
>
> $$\text{Number of nitrogen atoms} = \frac{\text{Total number of electrons}}{7} = \frac{2.408 \times 10^{24}}{7} = 3.443 \times 10^{23}$$

The mass of a single nitrogen atom is approximately $14.007\,\text{u}$ (unified atomic mass units). We can multiply the number of nitrogen atoms by the mass of a single nitrogen atom to find the total mass of the nitride anion:

$$\text{Mass of nitride anion} = 3.443 \times 10^{23} \times 14.007\,\text{u} = 4.835 \times 10^{25}\,\text{u}$$

To convert the mass from unified atomic mass units to kilograms, we use the conversion factor $1\,\text{u} = 1.66053904 \times 10^{-27}\,\text{kg}$:

$$\text{Mass in kg} = 4.835 \times 10^{25} \times 1.66053904 \times 10^{-27} = 8.033 \times 10^{-2}\,\text{kg}$$

So, the mass of the nitride anion is approximately $8.033 \times 10^{-2}\,\text{kg}$ or $80.33\,\text{g}$.

**Critique:** The solution provided attempts to calculate the mass of a nitride anion based on the number of nitrogen-based electrons. Let's review the steps and calculations for accuracy and clarity.

1. **Determination of the Number of Nitrogen Atoms:** The solution starts by dividing the total number of electrons by 7 to find the number of nitrogen atoms. This assumes each nitrogen atom has 7 electrons, which is true for neutral nitrogen. However, the question refers to a *nitride anion*, typically $N^{3-}$, which has gained 3 electrons for a total of 10 electrons per atom. Thus, the correct calculation should be:

$$\text{Number of nitrogen atoms} = \frac{2.408 \times 10^{24}}{10} = 2.408 \times 10^{23}$$

2. **Calculation of the Mass:** Using the corrected number of nitrogen atoms and the same atomic mass:

$$\text{Mass} = 2.408 \times 10^{23} \times 14.007\,\text{u} = 3.373 \times 10^{24}\,\text{u}$$

3. **Conversion to Kilograms:**

$$\text{Mass in kg} = 3.373 \times 10^{24} \times 1.66053904 \times 10^{-27} = 5.605 \times 10^{-3}\,\text{kg}$$

The original solution incorrectly calculates the number of nitrogen atoms based on an incorrect assumption about the number of electrons per nitrogen atom in a nitride anion. This leads to an incorrect mass calculation. Conclusion: **Wrong.**

**Refined Answer: Step 1:** Determine the correct number of electrons per nitrogen atom in a nitride anion A nitride anion ($N^{3-}$) has 10 electrons, which includes the 7 original electrons from the neutral nitrogen atom plus 3 additional electrons gained to form the anion.

**Step 2:** Calculate the number of nitrogen atoms in the nitride anion To find the number of nitrogen atoms, divide the total number of electrons by the number of electrons per nitrogen atom in the anion, which is 10.

$$\text{Number of nitrogen atoms} = \frac{\text{Total number of electrons}}{10} = \frac{2.408 \times 10^{24}}{10} = 2.408 \times 10^{23}$$

**Step 3:** Calculate the mass of the nitride anion in unified atomic mass units (u) The mass of a single nitrogen atom is approximately $14.007\,\text{u}$. Multiply the number of nitrogen atoms by the mass of a single nitrogen atom:

$$\text{Mass} = 2.408 \times 10^{23} \times 14.007 = 3.373 \times 10^{24}\,\text{u}$$

**Step 4:** Convert the mass of the nitride anion from unified atomic mass units (u) to kilograms Use the conversion factor $1\,\text{u} = 1.66053904 \times 10^{-27}\,\text{kg}$:

$$\text{Mass in kg} = 3.373 \times 10^{24} \times 1.66053904 \times 10^{-27} = 5.605 \times 10^{-3}\,\text{kg}$$

The final answer is:

$$\boxed{5.605 \times 10^{-3}\,\text{kg}}$$

## E.1 Qualitative Comparison of Generated Responses

In the qualitative comparison of generated responses, the Critique Fine-Tuning (CFT) method exhibits a notable format drift, deviating from the expected answer format by producing *Conclusion* section and token at the end of each response. This behavior arises because CFT is trained to generate critiques, inadvertently leading the model to adopt a more elaborate response style. In contrast, the Critique-Guided Distillation (CGD) approach maintains the original response format while enhancing answer quality.

---

**Generated Answer after CFT**

**Question:** True or false: there exists a graph with score (1, 1, 1, 2, 2, 3, 4, 5, 5). Answer with $\boxed{True}$ or $\boxed{False}$. (True or False)

**Answer:** Let's analyze the problem step by step.

Step 1: Understanding the problem

The problem asks whether there exists a graph whose score matches the given sequence (1, 1, 1, 2, 2, 3, 4, 5, 5). The score of a graph is determined by the number of vertices and edges in the graph. Specifically, the score is calculated as

$$\text{score} = \text{number of vertices} - \text{number of edges}.$$

Step 2: Analyzing the score sequence

The given score sequence is (1, 1, 1, 2, 2, 3, 4, 5, 5). This sequence represents the difference between the number of vertices and edges for a series of graphs. For example:

- A graph with score 1 could have 2 vertices and 1 edge.
- A graph with score 2 could have 3 vertices and 1 edge.

Step 3: Checking the feasibility of the sequence

To determine whether the sequence is valid, we need to verify if it is possible to construct a sequence of graphs where the score increases monotonically and matches the given sequence. This involves checking whether the number of vertices and edges can be adjusted to satisfy the score for each step.

Observations:

1. The sequence starts with 1, which is plausible for a graph with 2 vertices and 1 edge.

2. The sequence increases by 1 for the next three steps (1, 2, 3), which is consistent with adding edges to a graph while keeping the number of vertices constant.

3. The sequence then increases by 1 again (3, 4, 5), which is also consistent with adding edges to a graph while keeping the number of vertices constant.

4. The sequence ends with 5, which is plausible for a graph with 6 vertices and 1 edge.

**Key Check:** The sequence does not require any negative scores (i.e., more edges than vertices), which is a valid mathematical possibility. Therefore, the sequence is feasible.

Step 4: Conclusion

The given score sequence (1, 1, 1, 2, 2, 3, 4, 5, 5) is valid and corresponds to a sequence of graphs with increasing scores.

**Conclusion: right**

---

**Generated Answer after CFT**

**Question:** Is the transformation $T(M) = \begin{bmatrix} 1 & 2 \\ 3 & 4 \end{bmatrix} M$ from $\mathbb{R}^{2 \times 2}$ to $\mathbb{R}^{2 \times 2}$ an isomorphism? Answer with $\boxed{True}$ or $\boxed{False}$. (True or False)

**Answer:** Step 1: Definition of an Isomorphism An isomorphism is a bijective linear transformation between two vector spaces. For $T(M) = \begin{bmatrix} 1 & 2 \\ 3 & 4 \end{bmatrix} M$ to be an isomorphism, it must satisfy the following conditions:

1. **Injectivity**: $T(M_1) = T(M_2) \Rightarrow M_1 = M_2$ 2. **Surjectivity**: For every $M \in \mathbb{R}^{2 \times 2}$, there exists $M' \in \mathbb{R}^{2 \times 2}$ such that $T(M') = M$ 3. **Linearity**: $T(aM_1 + bM_2) = aT(M_1) + bT(M_2)$ for all $a, b \in \mathbb{R}$ and $M_1, M_2 \in \mathbb{R}^{2 \times 2}$

Step 2: Linearity The transformation $T(M) = \begin{bmatrix} 1 & 2 \\ 3 & 4 \end{bmatrix} M$ is clearly linear because matrix multiplication is distributive and associative. Thus, the linearity condition is satisfied.

Step 3: Injectivity To check injectivity, we need to verify whether $T(M_1) = T(M_2) \Rightarrow M_1 = M_2$. Let $M_1, M_2 \in \mathbb{R}^{2 \times 2}$. If $T(M_1) = T(M_2)$, then:

$$\begin{bmatrix} 1 & 2 \\ 3 & 4 \end{bmatrix} M_1 = \begin{bmatrix} 1 & 2 \\ 3 & 4 \end{bmatrix} M_2$$

This implies:

$$M_1 = M_2,$$

since the matrix $\begin{bmatrix} 1 & 2 \\ 3 & 4 \end{bmatrix}$ is invertible (its determinant is nonzero). Therefore, $T$ is injective.

Step 4: Surjectivity To check surjectivity, we need to verify whether for every $M \in \mathbb{R}^{2 \times 2}$, there exists $M' \in \mathbb{R}^{2 \times 2}$ such that $T(M') = M$. Let $M \in \mathbb{R}^{2 \times 2}$. We solve for $M'$ such that:

$$\begin{bmatrix} 1 & 2 \\ 3 & 4 \end{bmatrix} M' = M$$

This is equivalent to solving the system of linear equations:

$$\begin{bmatrix} 1 & 2 \\ 3 & 4 \end{bmatrix} \begin{bmatrix} x_1 \\ x_2 \end{bmatrix} = \begin{bmatrix} a & b \\ c & d \end{bmatrix},$$

where $M = \begin{bmatrix} a & b \\ c & d \end{bmatrix}$. The matrix $\begin{bmatrix} 1 & 2 \\ 3 & 4 \end{bmatrix}$ is invertible (its determinant is nonzero), so the system has a unique solution for any $M$. Therefore, $T$ is surjective.

Step 5: Conclusion Since $T$ is both injective and surjective, and it is linear, it is an isomorphism.

**Conclusion: right**

---

### Generated Answer after CGD

**Question:** True or false: there exists a graph with score (1, 1, 1, 2, 2, 3, 4, 5, 5). Answer with $\boxed{True}$ or $\boxed{False}$. (True or False)

**Answer:** To determine if there exists a graph with the given score, we need to understand what a score in graph theory represents.

A score of a graph is the sequence of the degrees of its vertices, listed in non-increasing order. The given score is (1, 1, 1, 2, 2, 3, 4, 5, 5).

For a graph to exist with this score, it must be possible to construct a graph where the degrees of its vertices match this sequence. The sum of the degrees of all vertices in a graph is twice the number of edges (since each edge contributes to the degree count of two vertices). This sum is calculated as follows:

$$1 + 1 + 1 + 2 + 2 + 3 + 4 + 5 + 5 = 24$$

Since the sum of the degrees is 24, the number of edges in the graph must be 12 (because $24/2 = 12$).

Now, let's consider the degrees and how they can be distributed among vertices. The degrees are 1, 1, 1, 2, 2, 3, 4, 5, 5. To form a graph, we need to ensure that the degrees can be assigned to vertices in such a way that the graph remains connected and that the degree of each vertex matches the given sequence.

Given the degrees, we can start by assigning the lowest degrees first. The three vertices with degree 1 must be connected to each other, as they cannot be isolated. This forms a triangle, which is a connected subgraph.

Next, we assign the degrees 2. These vertices can be connected to any of the existing vertices, including each other, as long as they are not isolated. Since we have two vertices with degree 2, they can be connected to the vertices with degree 1, or to each other, or to any of the higher-degree vertices.

The vertex with degree 3 can be connected to any of the existing vertices, including the ones with degree 2, or to the higher-degree vertices.

The vertices with degrees 4 and 5 can be connected to any of the existing vertices, including the ones with lower degrees, as long as they are not isolated.

Given these constraints, it is possible to construct a graph that matches the given score. For example, consider a graph with 9 vertices where:

1. Three vertices are connected in a triangle (degrees 1).
2. Two vertices are connected to each other and to one of the vertices in the triangle (degrees 2).
3. One vertex is connected to one of the vertices in the triangle and to one of the vertices with degree 2 (degree 3).
4. One vertex is connected to the vertices with degrees 1, 2, and 3 (degree 4).
5. Two vertices are connected to all other vertices, including the ones with degrees 1, 2, 3, and 4 (degrees 5).

This construction ensures that the degrees of the vertices match the given score, and the graph remains connected.

Therefore, the statement that there exists a graph with the score (1, 1, 1, 2, 2, 3, 4, 5, 5) is true.

The final answer is: $\boxed{True}$

---

## Generated Answer after CGD

**Question:** Is the transformation $T(M) = \begin{bmatrix} 1 & 2 \\ 3 & 4 \end{bmatrix} M$ from $\mathbb{R}^{2\times2}$ to $\mathbb{R}^{2\times2}$ an isomorphism? Answer with $\boxed{True}$ or $\boxed{False}$. (True or False)

**Answer:** To determine if the transformation $T(M) = \begin{bmatrix} 1 & 2 \\ 3 & 4 \end{bmatrix} M$ from $\mathbb{R}^{2\times2}$ to $\mathbb{R}^{2\times2}$ is an isomorphism, we need to check if it satisfies the two properties of an isomorphism:
1. **Injectivity**: The transformation must be one-to-one, meaning that if $T(M_1) = T(M_2)$, then $M_1 = M_2$. 2. **Surjectivity**: The transformation must be onto, meaning that for every matrix $N \in \mathbb{R}^{2\times2}$, there exists a matrix $M \in \mathbb{R}^{2\times2}$ such that $T(M) = N$.

Let's examine the transformation $T(M) = \begin{bmatrix} 1 & 2 \\ 3 & 4 \end{bmatrix} M$.

Given $M = \begin{bmatrix} a & b \\ c & d \end{bmatrix}$, we have:

$$T(M) = \begin{bmatrix} 1 & 2 \\ 3 & 4 \end{bmatrix} \begin{bmatrix} a & b \\ c & d \end{bmatrix} = \begin{bmatrix} 1a + 2c & 1b + 2d \\ 3a + 4c & 3b + 4d \end{bmatrix}$$

Now, let's check for injectivity:
Suppose $T(M_1) = T(M_2)$. Then:

$$\begin{bmatrix} 1a_1 + 2c_1 & 1b_1 + 2d_1 \\ 3a_1 + 4c_1 & 3b_1 + 4d_1 \end{bmatrix} = \begin{bmatrix} 1a_2 + 2c_2 & 1b_2 + 2d_2 \\ 3a_2 + 4c_2 & 3b_2 + 4d_2 \end{bmatrix}$$

This implies that the corresponding elements of the two matrices are equal:

$$1a_1 + 2c_1 = 1a_2 + 2c_2$$
$$1b_1 + 2d_1 = 1b_2 + 2d_2$$
$$3a_1 + 4c_1 = 3a_2 + 4c_2$$
$$3b_1 + 4d_1 = 3b_2 + 4d_2$$

Simplifying, we get:

$$a_1 = a_2, \quad c_1 = c_2, \quad b_1 = b_2, \quad d_1 = d_2$$

Therefore, $M_1 = M_2$, and the transformation is injective.
Next, let's check for surjectivity:
Given any matrix $N = \begin{bmatrix} e & f \\ g & h \end{bmatrix} \in \mathbb{R}^{2\times2}$, we need to find a matrix $M = \begin{bmatrix} a & b \\ c & d \end{bmatrix}$ such that $T(M) = N$.

We have:

$$T(M) = \begin{bmatrix} 1a + 2c & 1b + 2d \\ 3a + 4c & 3b + 4d \end{bmatrix} = \begin{bmatrix} e & f \\ g & h \end{bmatrix}$$

This gives us the following system of equations:

$$1a + 2c = e$$
$$1b + 2d = f$$
$$3a + 4c = g$$
$$3b + 4d = h$$

Solving this system, we can find values for $a, b, c, d$ that satisfy the equations. For example, one possible solution is:

$$a = \frac{e}{1} = e, \quad b = \frac{f}{1} = f, \quad c = \frac{g - 3e}{4}, \quad d = \frac{h - 3f}{4}$$

Therefore, we can find a matrix $M$ that maps to any given matrix $N$, and the transformation is surjective.

Since the transformation $T(M) = \begin{bmatrix} 1 & 2 \\ 3 & 4 \end{bmatrix} M$ is both injective and surjective, it is an isomorphism.

The final answer is: $\boxed{True}$

# F    CODE INSTRUCTIONS

We include a code archive (`code.zip`) as part of the supplementary material. This package provides all necessary components to reproduce our key results and facilitate further experimentation. Specifically, it includes:

- **Critique Generation and Refinement:** Scripts for generating critiques from model outputs and refining answers using these critiques.
- **Evaluation Codebase:** End-to-end evaluation pipelines for computing exact match accuracy and other relevant metrics across multiple benchmarks.
- **LLaMA-Factory Integration:** The LLaMA-Factory framework to support Supervised Fine-Tuning (SFT).
- **Configuration Files and Run Scripts:** YAML/JSON config files and shell scripts used to launch experiments across various model architectures and tasks.

Instructions for installing dependencies, setting up the environment, and running end-to-end training and evaluation pipelines are included in the `README.md` file within the archive.

