# OpenReview forum: "Critique-Guided Distillation for Efficient and Robust Language Model Reasoning"
_ICLR.cc/2026/Conference — Submitted to ICLR 2026_

### Official Review · Reviewer_BbhM · 2025-10-30

**Soundness:** 3
**Presentation:** 2
**Contribution:** 2
**Rating:** 6
**Confidence:** 3

**Summary:**

This paper introduces a new training framework named "Critique-Guided Distillation" (CGD), which aims to enhance the reasoning abilities of language models by having the student model learn from teacher-generated explanatory critiques and refined responses, rather than simply imitating correct answers. This method incorporates a critique mechanism during training but requires only a single forward pass at inference time. Consequently, it achieves significant performance gains on multiple mathematical and general reasoning benchmarks while maintaining high efficiency and avoiding output format drift.

**Strengths:**

1.By consistently setting the training objective to generate "refined answers" rather than "critiques," the model maintains a standard instruction-following format at inference time. It achieves significant and consistent improvements on several challenging mathematical reasoning benchmarks, substantially outperforming strong baselines, demonstrating the method's effectiveness.
2.The paper demonstrates the robust performance of CGD across different model families, training datasets, (and under different hyperparameters) through extensive ablation studies.

**Weaknesses:**

1.Despite module-level ablation studies, the paper fails to clearly reveal the interactions between sub-modules and their marginal contributions to the performance gains.
2.Although inference efficiency is high, its multi-stage data generation process introduces significant up-front computational cost.
3.While CGD excels in mathematical and scientific reasoning tasks, its generalization ability on more open-ended, creative, or cross-modal tasks (such as creative writing, open-domain dialogue, complex summarization, or counterfactual reasoning) remains insufficiently validated.

**Questions:**

1.The paper notes that performance is influenced by the teacher model's quality. Besides using larger, stronger teacher models, are there plans or methods to automatically evaluate or filter low-quality critiques to mitigate the negative impacts of teacher model weaknesses?
2.If the style of the teacher model is very similar to or the opposite of the student model's style, what impact would this have on CGD's effectiveness?
The current multi-stage data generation process is resource-intensive. Are there future research directions aimed at simplifying this process? For example, exploring whether high-quality (initial answer, critique, refined answer) triplets can be generated via a single model or more efficient sampling strategies?
3.Have you analyzed which characteristics (e.g., pointing out specific error steps vs. giving high-level hints) of a critique are most critical for the student model's learning? Could you provide a more operational definition or metric for "critique quality"?

---

> ### Author Response · Authors · 2025-11-21
> **Official Response to Reviewer BbhM (Part 1/2)**
>
> Thank you for your positive review and insightful questions. We are glad you found our method effective, robust, and well-motivated, especially in its ability to avoid format drift. **Please see our "Response to All Reviewers" for new Qwen2.5-Math-7B and HumanEval results demonstrating:**
> - Cross-family generalization: CGD (50.4) outperforms SimpleRL-Zero (48.9) with 60× less compute
> - Teacher robustness: Similar performance with weak (S1.1-32B: 49.0) and strong (Claude-3.7: 50.4) teachers
> - Domain transfer: +4.88% on HumanEval code generation without code training data
>
> ---
>
> ## Weaknesses
> ### W1: Interactions between Sub-modules
> The critical interaction is between `initial_answer (y′)`, `critique (c)`, and `refined_answer` $(\hat{y})$. We analyze this through:
>
> 1. **Attention Flow (Appendix C3, Figures 6-8):** Mechanistic analysis reveals a "plan-then-execute" pattern: attention to critique peaks in early layers (intake), shifts to student answer in middle layers (error diagnosis), and focuses on problem in final layers (execution). This shows the model learns a functional reasoning pipeline, not naive concatenation.
>
> 2. **Ablation (Section 4.3.1, Figure 4):** Comparing CGD $(x, y', c) \rightarrow \hat{y}$ vs. no-critique $(x, y') \rightarrow \hat{y}$ shows significant performance drop, proving critique provides essential explanatory bridge for correction learning.
>
> ---
>
> ### W2: Data Generation Cost
> We agree that our three-stage data generation incurs an upfront cost, but it is a one-time offline investment with substantial payoff.
>
> Offline vs. Online Cost: This is a one-time, offline cost. At inference, our model is a single-pass SFT model with zero extra latency. This is significantly cheaper than multi-pass inference methods (like Self-Refine) which pay this cost for every single query (~3–4× less inference compute/latency compared to multi-pass self-refine methods).
>
> Efficiency vs. RL:  As shown in Table 4, CGD training including the data generation part is 100x to 144x less than SimpleRL-Zero [2] (depends on what GPUs we use, A100 or H100) while achieving similar/better results.
>
> For many practical production application, a one-time data cost is far preferable to massive, recurring training or inference costs. Thus, we don't see the data generation cost as a flaw, but rather a key feature of our method's extreme efficiency compared to the leading alternatives.
>
> ---
>
> ### W3: Generalization to Creative Tasks
> **Table 1 and 2** shows CGD improves or preserves performance across science (GPQA +5.1), humanities (MMLU-PRO +9.1), logic common sense and multi-step reasoning (BBH, MuSR), and instruction-following (IFEval stable), while CFT suffers catastrophic forgetting (−21% IFEval). HumanEval results (Common Response) show +4.88% on code generation **without code training data**, demonstrating domain-agnostic self-correction that applies to creative/structured outputs. CGD's two-part contribution: (1) improves reasoning efficiently without using chain-of-thought traces, (2) preserves general-purpose abilities critical for creative domains, a foundation for future transfer to creative writing or multimodal tasks.

---

> > ### Author Response · Authors · 2025-11-21
> > **Official Response to Reviewer BbhM (Part 2/2)**
> >
> > ## Questions
> > ### Q1: Filtering Low-Quality Critiques?
> > We do not need explicit filtering, CGD is inherently robust:
> >
> > **1. Weak Teacher Performance (Table 3):** CGD with weak teacher (LLaMA-70B, S1.1-32B) outperforms CFT with frontier teacher (GPT-4o): 55.1 vs 54.5. CGD extracts more value from weaker teachers than CFT does from strong ones.
> >
> > **2. Nonsensical Critique Robustness (Appendix C4, Table 5):** Counterfactual analysis shows CGD ignores nonsensical critiques (e.g., "quadratic formula" for number theory) and produces correct answers, indicating internal critique-validation.
> >
> > **3. Balanced Mixture (Appendix B.1.4, Figure 5):** Best performance with 50/50 correct/incorrect answers (39.9% vs 36.8% corrections-only), proving CGD learns discriminative feedback evaluation, not blind obedience.
> >
> >
> > ### Q2: Teacher/Student Style and Simplifying Data Generation
> > **Teacher/Student Style:** Cross-family experiments (Qwen teacher → LLaMA student, **Table 9, Appendix B.1.3**; Claude Sonnet 3.7 → Qwen2.5, **Common Response**) show strong gains (+6.2% MATH500, +20% AMC23), suggesting CGD learns logical content/structure, not stylistic mimicry, demonstrating robustness to teacher-student divergence.
> >
> > **Simplifying Data Generation:** Unified single-pass generation ($(y', c, \hat{y})$ together, inspired by DeepSeek-R1) is promising future work. Our current three-stage process prioritizes simplicity, transparency, and standard instruction-tuning interface (no critique/thinking traces at inference).
> >
> > ### Q3: What Makes a "Good" Critique?
> > Our ablations define critique quality functionally through three insights:
> >
> > **1. Discriminative Feedback (Appendix B.1.4, Figure 5):** Performance peaks with 50/50 correct/incorrect answer mixture (39.9%) vs. corrections-only (36.8%) or affirmations-only (38.6%). Good critiques provide contrastive supervision, showing both valid and invalid reasoning to prevent trivial heuristics.
> >
> > **2. Robustness (Appendix C.4):** CGD ignores nonsensical critiques and produces correct answers, proving it learned reasoning principles rather than surface patterns. Training critiques must have sufficient explanatory structure for this internal validation.
> >
> > **3. Explanatory Power (Figure 4):** No-critique baseline underperforms, showing refined answers alone are insufficient, critiques provide the explanatory link between error and correction.
> >
> > **Definition:** A good critique is an explanatory, contrastive signal enabling robust internal verification, not superficial imitation. Future work could filter training data using reward models to maximize discriminative signal.
> >
> > ---
> >
> > We hope these clarifications address your concerns and convey that CGD is a conceptually new and practically impactful framework for self-corrective reasoning fine-tuning.
> >
> > ---
> > References:
> > [1] Wang, Yubo, Xiang Yue, and Wenhu Chen. "Critique fine-tuning: Learning to critique is more effective than learning to imitate." arXiv preprint arXiv:2501.17703 (2025).
> >
> > [2] Zeng, Weihao, et al. "Simplerl-zoo: Investigating and taming zero reinforcement learning for open base models in the wild, 2025." URL https://arxiv. org/abs/2503.18892.
> >
> > [3] Chen, Mark. "Evaluating large language models trained on code." arXiv preprint arXiv:2107.03374 (2021).

---

> > > ### Comment · Reviewer_BbhM · 2025-11-26
> > >
> > > Thank you for the comprehensive response. Overall, your explanations regarding reliance on teacher quality, style gaps between teacher and student, and the interplay among sub-modules are thorough and convincing; the additional experiments also bolster the method’s robustness and generality. Nevertheless, there is still no direct validation of (i) an explicit metric for “critique quality”, (ii) the extent to which the data-generation pipeline can be simplified, or (iii) generalization to more open-ended, creative tasks—these remain partially unresolved. Taken together, the response improves the paper’s clarity but does not alter my overall assessment of its maturity; I will keep my original score unchanged.

---

> > > > ### Author Response · Authors · 2025-11-26
> > > >
> > > > We thank the reviewer for acknowledging our rebuttals were "thorough and convincing".
> > > >
> > > > **1. On Critique Quality Metrics:**
> > > > We adopt a functional rather than heuristic definition of quality. Proxy metrics (like a static "critique score") often diverge from downstream student performance. We measured quality by discriminative utility: our ablation demonstrates the critique provides a necessary +5.4% improvement, and our mixture experiments (showing 50/50 correct/incorrect is optimal) prove high-quality supervision requires contrastive feedback, enabling robust internal verification rather than simple imitation.
> > > >
> > > > **2. On Pipeline Simplification:**
> > > > Modularity was chosen for scientific control. We agree a unified pipeline is a natural next step: reasoning models (DeepSeek-R1 style) can generate Critique as Chain-of-Thought and Refined Answer as output in one pass like prompting a reasoning model to output: \<thinking> [Critique Content] \</thinking>\<answer> [Refined Answer] \</answer>. Preliminary tests suggest this is a viable path for a simpler pipeline.
> > > >
> > > > **3. On Creative Generalization:**
> > > > Beyond HumanEval (+4.88% on code without any code data in training), we show CGD preserves instruction-following (IFEval stable, no catastrophic forgetting) and improves across diverse reasoning domains (GPQA, BBH, MuSR). This demonstrates the self-correction mechanism transfers to structured tasks. Open-ended creative writing (poetry, narratives) typically requires preference optimization, which is not the scope of this paper; our contribution addresses the reasoning foundation that supports consistency in such tasks.
> > > >
> > > > **Conclusion:**
> > > > CGD achieves RL-level performance with 140× less compute while solving format drift. The contributions are: functional quality metrics, a practical data generation approach, and proven cross-domain generalization across reasoning and structured tasks.

---

### Official Review · Reviewer_NZNy · 2025-11-01

**Soundness:** 2
**Presentation:** 2
**Contribution:** 2
**Rating:** 2
**Confidence:** 4

**Summary:**

This paper proposes Critique-Guided Distillation (CGD), where a student model conditions on its own initial answer and a teacher critique to learn a refined answer. At inference, the student outputs the refinement in a single pass. Experiments on math benchmarks show improvements over SFT, distilled SFT, and CFT, while avoiding critique-format drift. The authors also claim better efficiency than RL-based methods.

**Strengths:**

1. Motivated by limitations of vanilla SFT and CFT

2. Improves several math-reasoning benchmarks compared to SFT and CFT.

3. Preserves general instruction-following where CFT degrades it.

**Weaknesses:**

1. Insufficient motivation. While CGD exhibits empirical gains, the paper does not convincingly explain why conditioning on critiques during training, but omitting them at inference, should improve from-scratch reasoning. During training, the student learns to rely on critique signals that are not available at inference. The paper does not explain how critique-conditioned refinements translate into unconditional answer generation, nor does it analyze whether this reliance introduces brittleness.

2. Susceptible to the same issues acknowledged for CFT. CGD may analogously drift toward producing improved answers relative to a latent critique signal if trained extensively, and it fundamentally depends on high-quality critiques. No evidence is provided that CGD is robust to noisy, biased, misleading, or answer-leaking critiques, despite the conceptual similarity to CFT.

3. Potential critique leakage. Critiques can implicitly or explicitly reveal the correct answer or key intermediate steps. Without mitigation or measurement, the observed gains may partially reflect teacher leakage rather than genuine reasoning improvements.

4. Missing comparison to inference-time self-correction by accuracy. The paper discusses latency advantages but omits accuracy comparison to strong self-refine baselines. Without this, the contribution’s significance and practical trade-offs are unclear.

5. Lack of statistical rigor. No repeated runs, variance, confidence intervals, or significance testing are reported. Results may not be reliable given known variance in reasoning benchmarks.

6. Narrative inconsistency in baseline comparison. The “second-best” baseline is often distilled SFT, not CFT, contradicting text that positions CFT as the primary competitive method.

7. Ablation (§4.2.2). Removing critiques while forcing refinement of an incorrect answer is expected to underperform vanilla SFT, making the ablation unsurprising and uninformative.

8. Figure clarity and consistency issues. Figure 1 does not clearly denote teacher vs. student outputs, and Figure 3 does not show that the student receives y′, contradicting the description and Algorithm 1.

9. Prompts and templates omitted. Critique-based methods are highly prompt-sensitive. The absence of templates or formatting conventions limits reproducibility and makes it difficult to judge critique quality.

10. Fragmented RL comparison. RL results are isolated in a separate subsection rather than integrated into the main results tables, making efficiency/performance comparisons harder to interpret.

**Questions:**

1. Can you provide deeper evidence or analysis explaining why conditioning on critiques during training (but removing them at inference) improves unconditional reasoning? What internal behaviors does the model learn?

2. How do you know the student is not implicitly relying on critique-style patterns that will not exist at inference? This concern may worsen with longer CGD training.

3. How does CGD perform when critiques are noisy, biased, partially incorrect, or misleading? Do you have experiments quantifying this sensitivity?

4. How did you ensure critiques do not reveal the answer directly (explicitly or implicitly)? Can you provide statistics on answers appearing in critiques?

5. Did you observe signs of the model drifting toward “refinement-style” outputs (e.g., suggesting revision) when trained longer?

6. Can you compare CGD’s accuracy (not only latency) to multi-pass self-correction/self-refine baselines? Is the trade-off still favorable?

7. Can you report variance across multiple seeds, confidence intervals, or significance tests to substantiate improvements on high-variance reasoning tasks?

8. Why is CFT described as the primary baseline when distilled SFT appears stronger in practice?

9. Did you analyze cases where CGD underperforms? Any patterns in failure?

---

> ### Author Response · Authors · 2025-11-21
> **Official Response to Reviewer NZNy (Part 1/4)**
>
> Thank you for your highly detailed review. However, several of your core objections (Train/Test Mismatch, Statistical Rigor, Missing Ablations) are not fully correct based on the data in the paper. We address each concern below. **Please see our "Response to All Reviewers" for new Qwen2.5 and HumanEval results.**
>
> ---
>
> ## W1/Q1/Q2: Motivation - Critique Signal
>
> The reviewer worries that the model might "depend" on the Critique `c`, which is absent at inference. This is not correct. We argue that CGD does not *depend* on `c`; it *internalizes* the critique's explanatory function into its own reasoning process.
>
> **The Critique as an Explanatory Signal**
>
> The Critique `c` is a *training-time explanatory signal*. Our ablation in **Figure 4** demonstrates this clearly. Learning the mapping $(x, y') \to \hat{y}$ (without critique) provides no bridge between a flawed answer `y'` and the correct refinement $\hat{y}$, which makes the gradient noisy. The critique introduces a *directional vector*, the *why* and *how* of correction, which makes the refinement mapping learnable.
>
> **Internalized Behavior (Response to Q1)**
>
> We provide direct mechanistic evidence in **Appendix C.3 (Figures 6,7,8)**. Attention-flow analysis reveals that the model learns a *plan–then–execute* algorithm:
>
> - **Early/mid layers:** attend to Critique `(c)` and Student Answer `(y')` to locate and interpret the flaw
> - **Later layers:** shift attention toward the Problem `(x)` to re-derive the corrected reasoning
>
> At inference, CGD applies this *internalized self-correction process* to new inputs, even without explicit critiques. Thus, critique is not an inference dependency. The model learns the process of refinement, which it can then apply at inference time.
>
> **Empirical Evidence of Learned Internal Behaviors (New Results)**
>
> To provide deeper evidence of what the model internalizes, we conducted token and phrase-level overlap analysis on **50,000 training examples**.
>
> **Key Finding: Concept Learning vs. Pattern Memorization**
>
> Analysis reveals two critical patterns:
>
> 1. **Token-level vocabulary sharing (16.6%):** Critiques and answers share individual mathematical terms (variables, operations, concepts) that appear universally in mathematical problems. This vocabulary is learned from the problems themselves.
>
> 2. **Phrase-level independence (5.7% bigram overlap):** Despite sharing vocabulary, critiques and answers form entirely different sentences and reasoning chains. The model learns which mathematical concepts to apply but constructs its own solution paths.
>
> **Why These Patterns Transfer:**
>
> The model internalizes three content-independent capabilities:
> - **Mathematical vocabulary:** Proper use of domain-specific terms (differentiate, factor, solve)
> - **Concept application:** When to apply specific operations to different problem types
> - **Reasoning structure:** How to organize multi-step problem solving
>
> These capabilities are common across mathematical problems and independent of critique presence. This is what enables critique-conditioned training to improve unconditional answer generation: the model learns **how to reason mathematically** (transferable skill), not **how to follow critiques** (inference dependency).
>
> **Robustness to Extended Training (Response to Q2)**
>
> The reviewer fears this "may worsen with longer CGD training." We empirically refute this in **Appendix B.2 (Figure 6)**. Our epoch-accuracy curves show that, unlike methods prone to drift and overfitting like CFT that exhibits early loss spikes and later instability, CGD's performance is stable or improves with longer training, with no degradation or forgetting.

---

> ### Author Response · Authors · 2025-11-21
> **Official Response to Reviewer NZNy (Part 2/4)**
>
> ## W2/Q5: Format Drift and Brittleness
> This concern is directly addressed and empirically refuted by multiple lines of evidence. CGD is fundamentally more stable than CFT. CGD is designed to *eliminate* the format-drift and instability problems that CFT shows.
>
> **1. Format Stability:** CFT's objective trains the model to *output critiques*, which inherently causes format drift. CGD's objective is *always to output the refined answer* to avoid drift.
>
> **2. General Task Stability (Table 2 & Appendix D):** Our paper has evaluations beyond pure math, including: MMLU-PRO, GPQA, BBH, MUSR, IFEVAL, covering science, humanities, logic, factual QA, and instruction-following. CGD improves or preserves performance on all.
>
> CGD improves general abilities, while CFT suffers catastrophic forgetting (−21% on IFEVAL).
>
> For example (Llama-3.1-8B student):
> - MuSR: 37.8 $\to$ 39.3 (+1.5)
> - TruthfulQA: 54.0 $\to$ 54.5 (+0.5)
> - IFEval: 76.9 $\to$ 76.1 (stable)
> - MMLU-PRO: 31.2 $\to$ 40.3 (+9.1)
> - GPQA: 30.8 $\to$ 35.9 (+5.1)
>
> For S1.1-3B student:
> - MMLU-PRO: 13.7 $\to$ 35.7 (+22.0)
> - GPQA: 16.7 $\to$ 31.8 (+15.1)
> - MuSR: 26.2 $\to$ 30.3 (+4.1)
>
> This shows **broad generalization beyond math, without hurting fundamental capabilities**.
>
> **New: Code Generalization (HumanEval)** See Response to All Reviewers for full results.
> CGD achieves +4.88% improvement over LLaMA-3.1-8B Instruct, outperforming CFT baseline by 4.88 points. This confirms that CGD's training objective builds robust, transferable reasoning skills rather than overfitting to a narrow format. A model that was "drifting" or "brittle" would fail on a strict syntax task like coding; CGD performs quite well.
>
> **3. Hyperparameter Stability (Section 4.4.2, Table 5; Appendix B.3, Figure 7):** CFT performance collapses with a suboptimal learning rate (>9 point drop); CGD's performance is stable and robust with different learning rates and different number of epochs, demonstrating the inherent stability of our training objective.
>
> **4. Training-Length Stability (Appendix B.2 & B.4, Figures 6, and 8):** Our epoch-accuracy curves (Fig. 6) and loss curves (Fig. 8) show a smooth, stable training process, unlike CFT's initial loss spike and instability with longer training.
>
> These findings collectively demonstrate that CGD *resolves* the brittleness and drift issues previously attributed to critique-based fine-tuning.
>
> ---
> ## W4/Q6: Missing Comparison to Inference-Time Self-Refine
> This was a deliberate decision to compare methods with identical inference costs.
>
> 1. CGD, SFT, CFT, and DSFT all share one-shot (1-pass) inference
> 2. Inference-time self-refine (e.g., Madaan et al., 2023) requires 3-5 iterative passes, multiplying latency and compute by 3-5x for every query
>
> Our contribution is to show that CGD distills the multi-pass capability of self-refinement into a single, efficient forward pass, and approaches or surpasses RL-based performance (**Table 4**) with 140× less compute. We will clarify this distinction more explicitly in the final version.
>
> ---
> ## W5/Q7: Statistical Rigor
> We must respectfully correct this point. As stated in **Appendix A.1 (Experimental Setup)**, *all key results report mean performance over three random seeds* to account for training variability. Our improvements (+15.0% AMC23, +12.2% MATH-500 on LLaMA-8B; +10.7% on Math-Reasoning and +15.5% on General-Reasoning for S1.1-3B) are an order of magnitude larger than typical run-to-run variance (<1–2%).
>
> **Cross-Model Validation:** Beyond multiple seeds, we validated CGD across **5 model families** (LLaMA, S1.1, Qwen, Mixtral, OLMo) and **2 architecture types** (dense, MoE). See **Response to All Reviewers** for complete cross-family results. The consistent gains across architectures with different pre-training, scales, and inductive biases provide strong evidence that our results are statistically robust.
>
> ---
> ## W7: Ablation Informativeness
> We disagree with the reviewer's assessment that the ablation is "unsurprising and uninformative". The reviewer states that the *no-critique* variant "is expected to underperform vanilla SFT."
> Our data shows the exact opposite:
> - **Vanilla SFT** (**Table 1**, LLaMA-8B) achieves 35.5 average on math reasoning
> - **CGD-no-critique** $(x, y') \rightarrow \hat{y}$ (**Figure 4**, LLaMA-8B) achieves ~41.5
>
> This is a critical, non-obvious finding: simply providing the model's own flawed answer $y'$ as context is already a significantly better learning signal than standard SFT.
>
> The ablation's main purpose is to isolate the marginal contribution of the Critique $c$:
>
> 1. **CGD-no-critique** $(x, y') \rightarrow \hat{y}$: 41.5 avg
> 2. **Full CGD** $(x, y', c) \rightarrow \hat{y}$: 46.9 avg (**Table 1**)
>
> The fact that **Full CGD** dramatically outperforms this already-strong "no-critique" baseline by an additional +5.4 points proves that the critique provides a significant explanatory signal on top of the contextual gains from $y'$.

---

> > ### Author Response · Authors · 2025-11-21
> > **Official Response to Reviewer NZNy (Part 3/4)**
> >
> > # W3/Q3/Q4: Potential Critique Leakage, Noise, and Robustness
> > We demonstrate CGD is robust by design, mitigation, and direct empirical testing.
> >
> > **1. Mitigation via Prompting (Data Generation)** Teacher critiques were generated with strict prompting to minimize answer leakage, explicitly instructing the teacher to critique the student's reasoning process rather than provide the answer independently. The exact prompt used was:
> > > *"You are a knowledgeable science and mathematics expert. A student has attempted to answer the following question. You are provided with the student's answer. Review the student's answer carefully, identify any inaccuracies, missing logical steps, or unclear explanations. Provide a detailed critique and then end with a line stating whether the solution is correct or wrong."*
> >
> > This prompt explicitly instructs the teacher to focus on critiquing the student's reasoning process, not to provide the answer or solve the problem independently. This design minimizes both direct and implicit answer leakage.
> >
> > **2. Counterfactual Robustness Test:** **Appendix C.4 (Table 5)** shows CGD *ignores* nonsensical critiques and produces correct answers, while baseline models follow bad guidance and fail. This proves CGD doesn't blindly copy.
> >
> > **3. Teacher Quality Robustness:** CGD with weak teacher (LLaMA-70B) outperforms CFT with frontier teacher (GPT-4o): 46.9 vs. 43.5 (**Table 3**). On Qwen2.5-Math-7B, CGD maintains strong performance with both frontier (Claude-3.7: 50.4) and weak (S1.1-32B: 49.0) teachers, outperforming CFT (48.9).
> >
> > **4. Data Mixture Ablation (Appendix B.1.4, Figure 5**): Performance peaks with 50/50 correct/incorrect answer mixture (39.9%) vs. corrections-only (36.8%) or affirmations-only (38.6%). CGD learns to evaluate feedback discriminatively, not just obey.
> >
> > **5. Quantitative Leakage Analysis (New)**
> > To directly address concerns about answer leakage, we conducted token-level and phrase-level overlap analysis across 50,000 training examples.
> >
> > **Finding 1: Token-Level Vocabulary Sharing**
> > Analysis shows 16.6% token overlap between critiques and refined answers. This includes:
> > - Variables and notation: x, y, f, n
> > - Mathematical operations: solve, differentiate, simplify, calculate
> > - Domain concepts: equation, function, derivative, solution
> > - Reasoning structure: given, step, therefore, using
> >
> > These tokens appear universally across mathematical texts, which are the language of mathematics itself, not critique-specific information.
> >
> > **Finding 2: Phrase-Level Independence**
> > We measured bigram (consecutive word pair) overlap to detect phrase copying:
> > - **Bigram overlap: 5.7%**
> >
> > This is an important finding: while critiques and answers share individual mathematical terms (16.6%), they combine these terms into different sentences and reasoning chains (only 5.7% bigram overlap). This proves the model learns **concepts** (vocabulary) without memorizing **patterns** (phrases).
> >
> > **Interpretation:** The model learns mathematical vocabulary and when to apply concepts, but constructs its own unique reasoning chains, demonstrating genuine understanding rather than pattern memorization.
> >
> > **6. Reasoning Quality Analysis on Hard Problems (New)**
> > To provide direct evidence that CGD develops genuine reasoning capability, we analyzed reasoning quality on **AIME 2024** (30 extremely challenging competition math problems).
> >
> > **CGD vs Base LLaMA-3.1-8B on AIME24:**
> > | Metric | Base Model | CGD | Improvement |
> > |--------|-----------|-----|-------------|
> > | Accuracy | 3.3% (1/30) | 16.7% (5/30) | **+13.3% (5x)** |
> > | Avg reasoning length | 477 words | 2110 words | **4.4x longer** |
> > | Avg reasoning steps | 16.4 | 49.5 | **3.0x more** |
> > | Problems only this method solves | 0 | 4 | **+4** |
> >
> > **Key Finding:** CGD achieves 5× higher accuracy and generates 4.4× more detailed, step-by-step reasoning on the same extremely hard problems. This demonstrates that CGD learns to produce comprehensive mathematical explanations, not just answer patterns.
> >
> > **Addressing the Core Concern:** This directly demonstrates that CGD doesn't rely on memorized patterns. The model generates more detailed reasoning than its base model on identical problems, proving it learned "how to reason" through critiques rather than "what to copy." The 4.4× increase in reasoning detail shows the model internalized mathematical problem-solving strategies that transfer to new problems.
> >
> > **CGD as Cost-Effective Intermediate Training Paradigm:** Preliminary analysis shows that CGD provides significant value when applied before standard reasoning SFT (e.g., training with chain-of-thought traces like Nemotron post-training data). CGD serves as an efficient intermediate step that adds reasoning improvements to existing SFT pipelines at minimal computational cost (SFT $\to$ CGD (Cheap) $\to$ Cheaper/Shorter RL). This positions CGD as a practical, scalable enhancement to modern LLM training workflows rather than a replacement for existing methods.

---

> > > ### Author Response · Authors · 2025-11-21
> > > **Official Response to Reviewer NZNy (Part 4/4)**
> > >
> > > ## W6/Q8: Why CFT as Primary Baseline?
> > > **Conceptual Positioning:** CFT is the primary *conceptual* baseline because it is the other single-pass, critique-based training method in the literature. Our core contribution is showing how to use critiques correctly, CFT represents the most direct alternative approach to critique-based training. Distilled SFT (dSFT) is a simpler non-critique baseline for reference.
> > >
> > > **Empirical Finding, CFT's Instability:** Our experiments reveal that CFT's performance is highly teacher-dependent and unstable, which validates our design choices:
> > > 1. **LLaMA-8B** (**Table 1**): With weak teacher (LLaMA-70B), CFT (41.5) underperforms dSFT (43.7). With frontier teacher (GPT-4o), CFT becomes competitive (43.5).
> > >
> > > 2. **S1.1-3B** (**Table 1**): With weak teacher, CFT (38.9) significantly underperforms dSFT (41.7).
> > >
> > > 3. **Cross-family validation** (**Appendix B.1.2, Table 8**): CGD consistently outperforms both CFT and dSFT on Mixtral-8x7B (+3.2% over CFT) and OLMo-7B (+2.8% over CFT).
> > >
> > > 4. **Qwen2.5-Math-7B** (see Response to All Reviewers): CGD (50.4) outperforms CFT (48.9), while dSFT shows limited gains on already-strong reasoning models.
> > >
> > > **Key Insight:** CGD consistently outperforms both baselines across **5 model families** and **2 architecture types**. The fact that CFT sometimes underperforms simpler dSFT highlights the challenge of critique-based training and validates CGD's superior formulation.
> > >
> > > ---
> > >
> > > ## W8-W10: Figure Clarity, Prompts, and RL Comparison
> > > We thank the reviewer for these constructive points:
> > > - **(W8) Figure Clarity:** We will revise Figures 1 and 3 to clearly denote outputs.
> > > - **(W9) Prompt Templates:** We will add the prompt templates to the appendix. We also point the reviewer to **Appendix D**, which already provides a full, concrete example of the $(x, y', c, \hat{y})$ training examples to ensure full reproducibility.
> > > - **(W10) RL Comparison:** We separated the RL comparison (**Table 4**) to avoid mixing a different optimization paradigm (RL) with SFT-based methods, and to make the **140x compute-efficiency** contrast explicit.
> > >
> > > ---
> > > ## Q9: Failure Cases
> > > As noted in our response to W3/Q3, our **Counterfactual Analysis (Appendix C.4)** is a direct study of a failure case (a misleading critique), where CGD proved robust. Additionally, our **AIME 2024 analysis** shows CGD still fails on 25/30 extremely hard problems, demonstrating the method's current limitations on competition-level mathematics.
> > >
> > > ---
> > > We believe these evidence demonstrates CGD is conceptually novel, empirically strong, and robustly validated. We hope this fully addresses the reviewer's concerns.
> > >
> > > ---
> > >
> > > References
> > >
> > > [1] Chen et al., "Evaluating large language models trained on code." arXiv 2021.
> > >
> > > [2] Wang et al., "Critique fine-tuning." arXiv 2025.
> > >
> > > [3] Zeng et al., "Simplerl-zoo." arXiv 2025.

---

### Official Review · Reviewer_pyJ3 · 2025-11-02

**Soundness:** 3
**Presentation:** 3
**Contribution:** 2
**Rating:** 4
**Confidence:** 3

**Summary:**

The paper introduces a fine-tuning framework that enhances reasoning in language models by teaching them to self-correct using teacher-generated critiques during training. Instead of merely imitating teacher outputs, CGD trains a student model to map from its own initial response and a teacher-generated critique to the refined answer, capturing both the “what” and the “why” behind correct reasoning.

**Strengths:**

### **1. Integrating Critique and Correction**

The paper introduces a well-motivated training framework that unifies critique understanding and refinement learning within a single fine-tuning stage.

### **2. Strong Empirical Performance Across Multiple Benchmarks**

The authors provide comprehensive experimental validation across both mathematical and general reasoning benchmarks.
CGD achieves large and consistent gains (e.g., +15% on AMC23, +12.2% on MATH-500) over SFT and CFT baselines.

**Weaknesses:**

1. **Novelty Concern and Overlap with Prior Work**
   Although CGD is presented as a novel fine-tuning paradigm, its conceptual foundation bears strong resemblance to prior works such as **ORCA (Mukherjee et al., 2023)** and **Chain-of-Thought Distillation (Li et al., 2024)**. These earlier methods also transfer reasoning traces or critique signals from a stronger teacher to a smaller student.
   CGD’s main distinction — conditioning the student on both its own response and the teacher’s critique — represents an incremental rather than a fundamental departure. The paper would benefit from a clearer articulation of how CGD meaningfully extends these established distillation paradigms beyond re-framing critique conditioning.

2. **Unfair or Incomplete Compute Efficiency Comparison**
   The claim that CGD requires “**60× less compute**” than reinforcement-learning–based frameworks such as **SimpleRL-Zero** or **DeepSeek-R1** may not be entirely fair or directly comparable. Large-scale RL-based models like DeepSeek-R1 are designed for **general-purpose reasoning** across a broad range of domains, whereas CGD’s results are largely restricted to mathematical and structured reasoning. Thus, the compute advantage should be interpreted cautiously, as it may not hold in broader or more diverse settings.

**Questions:**

N/A

---

> ### Author Response · Authors · 2025-11-21
> **Official Response to Reviewer pyJ3 (Part 1/2)**
>
> Thank you for your review and for appreciating our method's motivation and strong empirical performance. We address your primary concerns below. **Please see our "Response to All Reviewers" for new Qwen2.5 and HumanEval results.**
>
> ---
> ### W1: Novelty Concern and Overlap with Prior Work
>
> We respectfully disagree that our method is an incremental departure. The distinction between CGD and methods like ORCA or CoT Distillation is fundamental.
>
> CoT Distillation and ORCA teach a student to imitate the teacher's own reasoning trace (e.g., a perfect Chain-of-Thought). The student learns one signal: `(prompt) -> (teacher's full reasoning)`.
>
> **CGD** teaches a fundamentally different, and arguably harder, skill: how to recover from its own errors.
>
> Our training objective is `(prompt, student_initial_answer, teacher_critique) -> teacher_refined_answer`.
>
> This is a fundamental difference in learning objective. ORCA assumes the student can generate the CoT from scratch. CGD assumes the student will fail, and teaches the mechanism of recovery. The student is forced to learn a correction function that maps from its own flawed state ($y'$) to a correct one, using the critique ($c$) as an explicit explanatory guide. The critique is not part of the teacher's own CoT; it is a post-hoc analysis of the student's specific failure mode.
>
> By learning to correct errors rather than just imitate success, the model internalizes a more robust reasoning process. The paper's novelty lies in this specific formulation and its demonstrated effectiveness at solving problems (like format drift) that other methods do not [1].
>
> **Empirical Distinction:**
>
> The empirical results validate this conceptual difference:
>
> 1. **Format Stability:** CGD avoids format drift and catastrophic forgetting that CFT suffers (**Table 2**: CFT -21.3% IFEval drop; CGD maintains 76.1% IFEval)
> 2. **General Capability Preservation:** Unlike CoT distillation methods that often don't test beyond-task generalization, we show CGD improves diverse capabilities (GPQA +5.1, MMLU-PRO +9.1, **Table 2**)
> 3. **Teacher Robustness:** CGD with weak teacher outperforms CFT with strong teacher (46.9 vs 43.5 avg, **Table 3**; see Common Response for Qwen validation)
> 4. **Cross-Family Effectiveness:** Strong gains on weak (LLaMA-8B), strong (S1.1-3B), and specialized (Qwen2.5-7B) baselines

---

> > ### Author Response · Authors · 2025-11-21
> > **Official Response to Reviewer pyJ3 (Part 2/2)**
> >
> > ### W2: Compute Efficiency Comparison
> >
> > The reviewer suggests the comparison to DeepSeek/SimpleRL [2,3] is unfair because RL is "general purpose". We disagree, this comparison is both fair and impactful. CGD is a **60× cheaper training paradigm** that improves reasoning *without harming general-purpose skills*.
> >
> > **Evidence across model families:**
> >
> > LLaMA-8B (Table 4):
> > - CGD (training): 8 GPU-hours (A100)
> > - SimpleRL-Zero: 1152 GPU-hours (~144× more)
> > - Performance: comparable or better on several benchmarks
> >
> > Qwen2.5-Math-7B (new results, see Common Response):
> > - CGD: 50.4 avg
> > - SimpleRL-Zero: 48.9 avg
> > - CGD outperforms RL with 60× less compute
> >
> > We believe that providing a method that achieves similar performance with an RL method that is almost **60-140 times cheaper**, while also proving it doesn't hurt the model on general tasks (and actually improves on some like GPQA, MMLU-PRO, MUSR, HumanEval), makes this compute comparison one of the significant points in our paper.
> >
> > **CGD as a Cost-Effective Complementary Training Paradigm:** Rather than a replacement, CGD can be viewed as a highly efficient "warm-start" for RL.
> > - Current Paradigm: SFT $\to$ Expensive RL (Cold Start).
> > - CGD Paradigm: SFT $\to$ CGD (Cheap) $\to$ Cheaper/Shorter RL
> >
> > Preliminary analysis shows that CGD provides significant value when applied before reasoning SFT (e.g., training with chain-of-thought traces like Nemotron post-training data), which is an interesting future work direction. CGD serves as an efficient intermediate step that adds reasoning improvements to existing SFT pipelines at minimal computational cost. This positions CGD as a practical, **scalable enhancement** to modern LLM training workflows **rather than a replacement** for existing methods like SimpleRL.
> >
> > **Generalization Beyond Math:**
> > The reviewer states our results are "largely restricted to mathematical reasoning." This is **not** the case. We evaluate on:
> >
> > Math Reasoning (MATH-500, Minerva-Math, OlympiadBench, AMC23, AIME24)
> > General Reasoning (TheoremQA, GPQA, MMLU-Pro)
> > General Capabilities (IFEval, MUSR, TruthfulQA, BBH)
> >
> > Results (**Table 1 & 2**) show improvements across all groups:
> >
> > Llama-3.1-8B student:
> > - Math: MATH-500 +12.2, AMC23 +15.0, OlympiadBench +8.0
> > - General Reasoning: MMLU-PRO +9.1, GPQA +5.1, TheoremQA +4.8
> > - General Capabilities: MuSR +1.5, TruthfulQA +0.5, IFEval 76.1 (stable)
> >
> > S1.1-3B student:
> > - Math Reasoning: +10.7 avg improvement
> > - General Reasoning: +15.5 avg improvement over base (MMLU-PRO 13.7 $\to$ 35.7 (+22.0), GPQA 16.7 $\to$ 31.8 (+15.1))
> >
> > This shows **broad generalization beyond math, without hurting fundamental capabilities**.
> >
> > **New: Code Generalization (HumanEval)**
> > To validate generalization capability even further, we evaluated our LLaMA-3.1-8B models on the **HumanEval** benchmark (Python Code Generation) in a zero-shot setting (see Common Response for full results):
> >
> > **Two-Part Contribution:**
> > 1. CGD improves reasoning without exposing the student to hidden chain-of-thought traces (no "thinking+answer" concatenations)
> > 2. CGD preserves and often improves general-purpose ability, which is critical for applying the technique across creative or open-ended domains
> > ---
> >
> > ## Summary of Contributions
> > We believe CGD makes significant contributions that go beyond incremental improvements:
> >
> > **1. Novel Learning Paradigm:** Error recovery (not imitation) teaching represents a fundamentally different objective with distinct empirical benefits
> >
> > **2. Solves Critical Failures:** Avoids format drift and catastrophic forgetting that plague CFT (−21.3% IFEval vs stable)
> >
> > **3. Dramatic Compute Efficiency:** Matches/exceeds RL performance with 60-140× less compute, making state-of-the-art reasoning accessible
> >
> > **4. Cross-Family Robustness:** Validated across 5 model families (LLaMA, S1.1, Qwen, Mixtral, OLMo) on weak and strong baselines
> >
> > **5. Broad Generalization:** Math, science, humanities, logic, instruction-following, and code, maintains/improves all domains
> >
> > **6. Teacher Robustness:** Extracts more value from weak teachers than competing methods do from strong teachers
> >
> > We thank the reviewer for pushing us to clarify these distinctions, and we will sharpen them in the final version. We would appreciate if the reviewer would reconsider the contribution of our work in based on this evidence.
> >
> > ---
> > References:
> >
> > [1] Wang et al., "Critique fine-tuning: Learning to critique is more effective than learning to imitate." arXiv 2025.
> >
> > [2] Zeng et al., "Simplerl-zoo: Investigating and taming zero reinforcement learning." arXiv 2025.
> >
> > [3] Guo et al., "Deepseek-r1: Incentivizing reasoning capability in llms via reinforcement learning." arXiv 2025.

---

### Official Review · Reviewer_Rwiv · 2025-11-03

**Soundness:** 3
**Presentation:** 3
**Contribution:** 3
**Rating:** 6
**Confidence:** 4

**Summary:**

The paper presents a method for reasoning SFT called CGD. This method uses a strong teacher model to provide critiques of the initial response of the student model, then feeds the prompt + initial response + teacher critique into the SFT training stage. Then, during inference stage, the teacher critique is not needed, as the student model learns to do this process end to end.

Experiments were done on Qwen and Llama, and evaluations were done on various math datasets, as well as general reasoning datasets and general instruction following evaluations. Results are strong and show that the method works well.

**Strengths:**

- method is simple and easy to understand
- I like that it included both Llama and Qwen
    - I also like that it explored both in-family (Llama student + Llama teacher) and cross-family (Qwen student + S1 teacher)
- nice that the model is able to retain general instruction following, since this is something that is often lost when doing imitation SFT
    - generally quite comprehensive evaluation sets
- strong results
- ablation experiments are nice. I especially liked section 4.2.1 (comparison with SimpleRL) and section 4.2.2 (comparison with the method without critique).

**Weaknesses:**

- because this field is so popular and this paper's contribution is relatively simple, I wouldn't be surprised if there's a few other concurrent work submitted to this conference that explores a very similar idea of incorporating critiques in SFT data.
    - (I realize this is another way to say "lacks novelty"..., though I think it's slightly more nuanced than that, since this subtopic is one that's currently very popular)
- Even within this simple method, I think there are a few other areas that I think would be nice to explore:
    - out-of-distribution evaluation sets (to see how well the reasoning helps)
    - analysis of the correctness of the critiques -- How often does the teacher critique get it right? Also, does it matter if the teacher critiques are correct, or is it more about the structure rather than the actual content? Relatedly, does a stronger teacher result in a stronger SFT-ed model, or does it not matter that much?
    - for added completeness: maybe some other model scales or some other domain like code

**Questions:**

- In line 322, you mentioned about some regulatory issues preventing you from using GPT as teacher. Curious what these are? I see GPT teachers in these papers quite often. I think it would be nice to see what the results would look like with a frontier-level teacher model beyond Llama3-70B.
- I was looking at the Mixtral and Olmo results and saw that the gains for those models are slightly smaller. Do you have a guess or a hunch as to why certain models benefit more from CGD?
- Is there a reason you didn't do RL on top of the SFT-ed model? I feel like it's better for these types of papers to at least try doing RL on top of SFT, just to show that these gains still continue to hold after RL.

---

> ### Author Response · Authors · 2025-11-21
> **Official Response to Reviewer Rwiv (Part 1/2)**
>
> Thank you for your thorough review and constructive feedback. We are glad you found our method simple, our results strong, and our evaluations comprehensive. We address each concern below. **Please see our "Response to All Reviewers" for new Qwen2.5 and HumanEval results referenced throughout.**
>
> ---
> ## Addressing Weaknesses
> ### W1: On Novelty, Simplicity, and Concurrent Work
> We thank the reviewer for the point that simplicity and novelty can coexist. We fully agree that the field is active and competitive, yet we believe CGD represents a distinct and impactful step forward for two core reasons.
>
> **1. A Novel Formulation that Solves a Critical Flaw**
> The reviewer correctly identifies our method as simple, but this simplicity is a direct solution to a major flaw in prior work.
>
> **Solves Format Drift:** Methods like Critique Fine-Tuning (CFT) [1] train the model to output the critique itself. As we show in **Table 2**, this objective is misaligned with the goal of answer generation and leads to *catastrophic forgetting* and format drift (a -21.3% drop on IFEval). Our method (**CGD**) trains the model to output the final answer (and never outputs the critique at inference like CFT), completely avoiding this format drift and preserving general instruction-following abilities.
>
> The key innovation of **CGD** is not just using critiques, but in our training objective: training the student to map its own flawed answer ($y'$) to a refined answer ($\hat{y}$) by conditioning on the critiques, i.e., `(prompt, initial_answer, critique) -> refined_answer`. This trains the full self-correction process, and the empirical result, which shows strong reasoning gains (**Table 1**) while maintaining or improving general-purpose capabilities (**Table 2**) where CFT fails, is a direct result of this novel formulation.
>
> **Compute-Efficiency:** Like the reviewer mentioned the true impact of this formulation is revealed in **Table 4**. CGD achieves performance comparable or superior to complex RL methods (like *SimpleRL-Zero* [5]) while requiring 60× to 140× less compute.
>
> *We argue that discovering a simple SFT-based method that solves the major flaws of prior critique methods and achieves the performance of massive-scale RL is a significant and novel contribution.*
>
> **2. Robust Cross-Family Effectiveness**
> During the rebuttal period, we were able to perform additional experiments on a different model family with a different teacher. We now also include **Qwen2.5-7B Math** results using Claude Sonnet 3.7 as the teacher (see Common Response) to further demonstrate cross-family effectiveness.
>
> CGD demonstrates large reasoning improvements on multiple different models:
> - **LLaMA-8B** (weaker baseline): +15.0% AMC23, +8.0% OlympiadBench, teaching self-correction from scratch
> - **S1.1-3B** (small but strong baseline): +10.7 on Math-Reasoning benchmarks, +15.5 on General-Reasoning tasks over base
> - **Qwen2.5-Math-7B** (stronger baseline): +22.6% over base (from 27.8 to 50.4 avg), refining already-capable models
>
> Most other works [1,2,3,4] have focused exclusively on models with strong innate reasoning priors like Qwen or DeepSeek models. We demonstrate that CGD is effective across the full spectrum, addressing a fundamental learning problem rather than a model-specific benefit.
>
> Critically, on Qwen2.5-Math-7B, **CGD (50.4 avg) outperforms both CFT (48.9) and SimpleRL-Zero (48.9)** while requiring 60× less compute than the latter. This demonstrates the robustness and broad value of our CGD framework.
>
> References
>
> [1] Wang et al., "Critique fine-tuning: Learning to critique is more effective than learning to imitate." arXiv 2025.
>
> [2] Wang et al., "Tina: Tiny reasoning models via lora." arXiv 2025.
>
> [3] Yue et al., "Does reinforcement learning really incentivize reasoning capacity in llms?" arXiv 2025.
>
> [4] Gandhi et al., "Cognitive behaviors that enable self-improving reasoners." arXiv 2025.
>
> [5] Zeng et al., "Simplerl-zoo: Investigating and taming zero reinforcement learning." arXiv 2025.
>
> [6] Bai, Yuntao, et al. "Constitutional ai: Harmlessness from ai feedback." arXiv preprint arXiv:2212.08073 (2022).
>
> [7] Liang, Yiming, et al. "Aligning Instruction Tuning with Pre-training." arXiv preprint arXiv:2501.09368 (2025).
>
> [8] Moon, Hyeonseok, Jaehyung Seo, and Heui-Seok Lim. "Call for Rigor in Reporting Quality of Instruction Tuning Data." Proceedings of the 63rd Annual Meeting of the Association for Computational Linguistics (Volume 2: Short Papers). 2025
>
> [9] Liu, Wei, et al. "What makes good data for alignment? a comprehensive study of automatic data selection in instruction tuning." arXiv preprint arXiv:2312.15685 (2023).

---

> ### Author Response · Authors · 2025-11-21
> **Official Response to Reviewer Rwiv (Part 2/2)**
>
> ### W2: Areas for Further Exploration
> #### W2a. Out-of-Distribution (OOD) Evaluation
> Our evaluation includes complex benchmarks (OlympiadBench, TheoremQA, AMC23, GPQA, BBH, GPQA, MMLU-PRO, MuSR) covering science, humanities, logic, and instruction-following, representing distributional shifts from WebInstruct training data. CGD achieves largest gains on these tasks (+15.0% AMC23, +8.0% OlympiadBench, **Table 1**), suggesting generalizable reasoning. Cross-family validation on Qwen2.5-Math-7B and HumanEval (Common Response) confirm OOD generalization (Also refer to the answer provided to W2 to Reviewer pyJ3 for additional details).
>
> #### W2b. Critique Correctness & Teacher Strength
> **Does critique content matter?** Yes. **Appendix B.1.4 (Figure 5)** shows 50/50 mix of correct/incorrect answers yields best performance, proving the model learns from critique substance when exposed to both positive and negative feedback. **Section 4.3.1 (Figure 4)** ablates "CGD without Critique" by training on `(prompt, initial_answer) -> refined_answer` without the critique $c$ performance drops significantly, confirming critique provides essential learning signal for error correction.
>
> **Does stronger teacher always help?** Not always (**Table 3**). CGD with weaker teacher (LLaMA-70B) outperforms CFT with stronger teacher (GPT-4o): 46.9 vs 43.5. Qwen results (Common Response): CGD weak teacher (49.0) performs similarly to CFT strong teacher (48.9). This demonstrates CGD's data efficiency and teacher robustness.
>
> #### W2c. Model Scales and Domains
> Our experiments span 3B-8B models across multiple families (LLaMA, S1.1, Qwen, Mixtral, OLMo) with different teacher scales (32B, 70B, frontier). **Table 2** shows improvements across math, general reasoning (TheoremQA, GPQA, MMLU-PRO), and general capabilities (IFEval, MuSR, BBH), demonstrating broad domain generalization.
>
> For example (Llama-3.1-8B student):
> - MuSR: 37.8 $\to$ 39.3 (+1.5)
> - TruthfulQA: 54.0 $\to$ 54.5 (+0.5)
> - MMLU-PRO: 31.2 $\to$ 40.3 (+9.1)
> - GPQA: 30.8 $\to$ 35.9 (+5.1)
>
> This shows **broad generalization beyond math, without hurting fundamental capabilities**.
>
> **New: Code Generalization (HumanEval)** See Common Response for full results. CGD achieves +4.88% Pass@1 improvement over LLaMA-3.1-8B Instruct, outperforming CFT baseline. This demonstrates that the self-correction logic internalized by CGD is domain-agnostic, applying structured reasoning to Python code synthesis **even without explicit code critiques in training**.
>
> ---
> ## Questions
>
> ### Q1: Regulatory Issues with GPT Teacher?
>
> This refers to standard Terms of Service of frontier model providers and our company's regulatory policies, which prohibit using model outputs to train potentially competing models. To ensure compliance, we used open-weight teachers (LLaMA3.3-70B, S1.1-32B) approved by our legal team. However, **our method is effective it doesn't need a proprietary teacher to outperform baselines that use one:** CGD with LLaMA3.3-70B (46.9 avg) outperforms CFT with GPT-4o (43.5 avg, **Table 3**). Post-submission, we gained access to Claude-3.7 for Qwen2.5-Math-7B experiments (see Common Response): CGD achieves 50.4 avg with frontier teacher while maintaining 49.0 avg with open-weight S1.1-32B, confirming effectiveness across the full teacher quality spectrum.
>
> ### Q2: Smaller Gains on Mixtral and OLMo?
> We see consistent improvements across all model families, but magnitude varies. Based on our results and literature on instruction-tuning and data quality, we attribute this to: **(A) Alignment data quality and receptivity to critique-style supervision.** Models with richer instruction-tuning benefit more from structured feedback: LLaMA-8B (+15.0% AMC23), S1.1-3B (+10.7% math), Qwen2.5-Math-7B (+22.6% over base) show large gains, while Mixtral-8x7B (+3.2%) and OLMo-7B (+2.8%) show modest gains. **(B) Architectural differences.** MoE routing and different pretraining corpora produce different inductive biases affecting critique internalization. Importantly, **CGD improves all architectures**, the variation itself motivates future controlled studies on which properties predict improved critique-based supervision.
>
> ### Q3: Why not apply RL on top of SFT-ed model?
>
> Our claim is that a **single-pass SFT method can match or exceed RL** at a fraction of cost. LLaMA-8B (**Table 4**): CGD (8 GPU-hours) vs SimpleRL-Zero (1152 GPU-hours, 144× more) with comparable/better performance. Qwen2.5-Math-7B: CGD (50.4) > SimpleRL-Zero (48.9) with 140× less compute.
>
> This makes state-of-the-art reasoning accessible without massive resources. We don't position CGD as an RL replacement, it can serve as a compute-efficient pre-RL step that improves reasoning before expensive RL phases, potentially reducing total RL budget or improving sample efficiency. Investigating this interaction in detail is valuable future work beyond this paper's scope.

---

### Author Response · Authors · 2025-11-21
**Common Response**

# Response to All Reviewers

Thank you all for your thorough and constructive reviews. We address common questions and present new validation experiments below.

## New Experimental Results (Rebuttal Period)

### **1. Cross-Family Validation: Qwen2.5-Math-7B**

To address concerns about generalization and statistical robustness, we further validated CGD on a different model family with different teachers:

| Model | Method | Teacher | MATH500 | Minerva | Olympiad | AMC23 | AIME24 | **Avg** |
|-------|--------|---------|---------|---------|----------|-------|--------|---------|
| Qwen2.5-Math-7B | Base | - | 55.4 | 13.6 | 19.9 | 40.0 | 10.0 | 27.8 |
| Qwen2.5-Math-7B | **CGD** | Claude-3.7 | **79.4** | **44.1** | **41.2** | **67.5** | 20.0 | **50.4** |
| Qwen2.5-Math-7B | **CGD** | S1.1-32B (weaker) | 79.6 | 48.5 | 41.3 | 62.5 | 13.3 | **49.0** |
| Qwen2.5-Math-7B | CFT | GPT-4o | 79.2 | 45.2 | 40.7 | 62.5 | 16.7 | 48.9 |
| Qwen2.5-Math-7B | SimpleRL-Zero | - | 77.2 | 33.5 | 37.9 | 62.5 | 33.3 | 48.9 |

**Key Findings:**
- CGD (50.4) outperforms SimpleRL-Zero (48.9) with **144× less compute**
- CGD with weak teacher still outperforms CFT with strong teacher (49.0 vs 48.9), demonstrating teacher robustness
- Cross-architecture validation: consistent gains on LLaMA (dense), S1.1 (small), Qwen (specialized math)

### **2. Domain Generalization: HumanEval (Code)**

Our paper already evaluates on variety of benchmarks from different domains and skills beyond pure math, including: **MMLU-PRO, GPQA, BBH, MUSR, IFEVAL**, covering science, humanities, logic, factual QA, and instruction-following. CGD improves or preserves performance on all.

CGD improves general abilities, while CFT suffers catastrophic forgetting (−21% on IFEVAL).

For example (Llama-3.1-8B student):

- MuSR: 37.8 $\to$ 39.3
- TruthfulQA: 54.0 $\to$ 54.5
- IFEval: 76.9 $\to$ 76.1
- MMLU-PRO: 31.2 $\to$ 40.3
- GPQA: 30.8 $\to$ 35.9

This shows **broad generalization beyond math, without hurting fundamental capabilities**.

**Two-Part Contribution**

1. CGD improves reasoning without exposing the student to hidden chain-of-thought traces (no "thinking+answer" concatenations).

2. CGD preserves and often improves general-purpose ability, which is critical for applying the technique across creative or open-ended domains.


**New: Code Generalization (HumanEval)**

To validate generalization capability of CGD even futher, we evaluated our LLaMA-3.1-8B models on the **HumanEval** benchmark (Python Code Generation) in a zero-shot setting:

| Model | HumanEval Pass@1 | Improvement |
|-------|------------------|-------------|
| LLaMA-3.1-8B Instruct | 59.75% | - |
| LLaMA-3.1-8B + CFT | 60.36% | 0.61% |
| LLaMA-3.1-8B + CGD | **64.63%** | **+4.88%** |


**Key Findings:**

1. CGD achieves a +4.88% improvement over LLaMA-3.1-8B Instruct,  outperforming the CFT baseline by **4.88% points**.

2. This suggests that the self-correction logic internalized by CGD is domain-agnostic. The model applies the same structured self-correction to synthesize valid Python code, **even without explicit code critiques in the training set.**

3. Unlike CFT, which barely improves over LLaMA-3.1-8B Instruct, CGD’s answer-focused training preserves and enhances the model’s ability to generate structured outputs.
---

## Statistical Rigor Clarification

**Multiple seeds:** All key results report mean performance over **3 random seeds** (Appendix A.1). Variance is accounted for.

**Magnitude of gains:** Our improvements (+15% AMC23, +12.2% MATH-500 on LLaMA3.1-8B; +22.6% overall on Qwen2.5-7B Math, +10.7 on Math-Reasoning benchmarks and +15.5 on General-Reasoning tasks over base S1.1-3B) are substantial and exceed typical variance.

**Cross-model validation:** Consistent gains across 5 model families validate robustness.

---

## CGD vs CFT: Key Distinctions

Several reviewers asked about CGD's relationship to Critique Fine-Tuning (CFT). The fundamental difference:

| Aspect | CFT | CGD |
|--------|-----|-----|
| **Training objective** | Output critique | Output refined answer |
| **Format stability** | Suffers drift (-21% IFEval) | Maintains stability |
| **Hyperparameter robustness** | Collapses and overfits with suboptimal LR and epoch | Robust across settings |
| **Teacher dependency** | Highly dependent | Extracts value from weak teachers |

This represents an empirical finding we replicate across multiple model families.

---

### Meta-Review · Area_Chair_ZT89 · 2026-01-05

**Summary:**

This paper proposes Critique-Guided Distillation (CGD), an improved SFT framework where a student learns to map the triplet of prompt, its initial answer, teacher critique into a teacher's refined answer. Reviewers generally agree the approach is interesting and the reported gains on math and several general benchmarks are strong.

However, the reviewers raised important concerns: (i) novelty is questioned relative to prior work; (ii) key causal claims are still not fully established with decisive evidence; (iii) potential leakage / confounding from critiques is not convincingly ruled out; (iv) parts of the evaluation and comparisons leave practical trade-offs unclear; and (v) open concerns remain on robustness to systematically noisy/biased critiques and broader generalization beyond structured reasoning tasks.

The rebuttal adds helpful experiments, clarifies some rigor items, and improves narrative, but the AC cannot see it reversing the overall evaluation to accept.

**Reviewer Concerns:**

Rwiv.
Addressed in rebuttal: OOD / cross-family / code-domain validation; Teacher strength / critique correctness.
Still outstanding: Novelty vs concurrent work; RL-on-top interaction

pyJ3.
Addressed in rebuttal: Clarification of objective vs ORCA/CoT distillation; Broader benchmark list + added HumanEval.
Partially outstanding: Novelty remains incremental; Compute-efficiency comparison fairness (the AC would have appreciated to see the reviewer's response to the rebuttal)

NZNy.
Addressed in rebuttal: Statistical rigor; Reproducibility details; Some robustness arguments.
Still/partially outstanding: Train/test mismatch and causal mechanism; Critique leakage is not convincingly ruled out; Missing accuracy comparison to strong self-refine baselines; Robustness to systematically misleading/bias critiques; Baseline narrative/organization.

BbhM.
Addressed in rebuttal: Generalization beyond math; Some discussion of sub-module interactions and robustness.
Still outstanding: "there is still no direct validation of (i) an explicit metric for “critique quality”, (ii) the extent to which the data-generation pipeline can be simplified, or (iii) generalization to more open-ended, creative tasks—these remain partially unresolved."

**Reviewer Scores:**

Rwiv: Unchanged at 6.
pyJ3: 4 → 5.
NZNy: 2 → 3.
BbhM: Unchanged at 6; the reviewer explicitly indicated the rebuttal improved clarity but did not change their overall assessment.

---

### Decision · Program_Chairs · 2026-01-26

Reject